# RETHINKING PARETO FRONTIER: ON THE OPTIMAL TRADE-OFFS IN FAIR CLASSIFICATION

**Junyi Chai, Shenyu Lu & Xiaoqian Wang** [*]
Elmore Family School of Electrical and Computer Engineering
Purdue University
West Lafayette, IN 47907, USA
`{chai28,lu876,joywang}@purdue.edu`

## ABSTRACT

Fairness has become an arising concern in machine learning with its prevalence in decision-making processes, and the trade-offs between various fairness notions and between fairness and accuracy has been empirically observed. However, the inheritance of such trade-offs, as well as the quantification of the best achievable trade-offs, i.e., the Pareto optimal trade-offs, under varied constraints on fairness notions has been rarely and improperly discussed. Owing to the sub-optimality of fairness interventions, existing work fails to provide informative characterization regarding these trade-offs. In light of existing limitations, in this work, we propose a reformulation of the model-specific (MS) Pareto optimal trade-off, where we frame it as convex optimization problems involving fairness notions and accuracy w.r.t. the confusion vector. Our formulation provides an efficient approximation of the best achievable accuracy under dynamic fairness constraints, and yields systematical analysis regarding the **fairness-accuracy trade-off**. Going beyond the discussion on fairness-accuracy trade-offs, we extend the discussion to the **trade-off between fairness notions**, which characterizes the impact of accuracy on the compatibility between fairness notions. Inspired by our reformulation, we propose a last-layer retraining framework with group-dependent bias, and we prove theoretically the superiority of our method over existing baselines. Experimental results demonstrate the effectiveness of our method in achieving better fairness-accuracy trade-off, and that our MS Pareto frontiers sufficiently quantify the two trade-offs.

## 1 INTRODUCTION

As machine learning becomes increasingly integrated into various societal domains, growing concerns have emerged regarding fairness issues (De-Arteaga et al., 2022; Barocas et al., 2023; Chen et al., 2023a), where models can potentially reflect or perpetuate real-world discrimination without proper regularization. Work has shown that fairness cannot be achieved simply through unawareness of sensitive information (Pessach & Shmueli, 2022; Mehrabi et al., 2021), owing to the implicit correlation between input features and sensitive information. Consequently, multiple fairness criteria and fairness interventions (Hort et al., 2022; Li et al., 2023a; Pessach & Shmueli, 2023) have been proposed to quantitatively measure and rectify the disparities between sensitive groups.

Empirical observations indicate that improvements in fairness may be accompanied by a deterioration in utility, leading to the fairness-accuracy trade-off (Zliobaite, 2015; Menon & Williamson, 2018; Zafar et al., 2019). It has been shown that there exists an inherent trade-off between demographic parity (DP) (Dwork et al., 2012) and accuracy, under the variations in base rates of different groups (Zhao & Gordon, 2022). Recent work demonstrates the intrinsic trade-off in fair representation learning, where the removal of sensitive information can be detrimental owing to its correlation with downstream tasks (Li et al., 2023b). Despite the insight of such theoretical analysis, it fails to quantify such phenomena under a dynamical setting, i.e., how the accuracy changes as the constraint on fairness changes. In addition, existing work has not yet theoretically characterized the trade-offs

---

[*]Corresponding author.

between fairness and accuracy for error-based fairness notions, including equal opportunity (EOp) and equalized odds (EOd) (Hardt et al., 2016). Dutta et al. (2020) suggests that there is no necessary trade-off between error-based fairness notions and accuracy; however, the analysis requires strong assumptions regarding data and the classifier, hindering the generalizability to real-world scenarios.

Furthermore, existing work has pointed to the tension in enforcing different fairness notions simultaneously, resulting in the trade-off between fairness notions. Different notions of fairness have been shown to conflict with each other, given the variations in base rates (Kleinberg et al., 2016; Chouldechova, 2017). Such conflict is also known as the 'impossibility results'. Going beyond the impossibility, more recent work (Reich & Vijaykumar, 2021; Gultchin et al., 2022) states the possibility of achieving various fairness notions concurrently, albeit with certain levels of fairness violations. Nonetheless, existing discussion has yet to quantitatively characterize the inherency of the trade-off between fairness notions. Moreover, while the correlation between such trade-offs and accuracy has been recognized in existing literature, it remains unclear how the trade-off changes as the accuracy values differ, impeding the discussion on balancing between fairness notions.

Following existing literature (Martinez et al., 2020; Wei & Niethammer, 2022), for a specific network structure, we seek to find the **best achievable accuracy** under different fairness notions and under varied fairness constraints, also referred to as the model-specific (MS) trade-off. For instance, when choosing ResNet-50 as baseline, the 'model-specific' refers to all fairness interventions applied to ResNet-50. We refer to Section 3 for a detailed discussion. This concept is also known as the **Pareto optimality** (Hochman & Rodgers, 1969), where no action or allocation is available that makes one individual better off without making another worse off. And the corresponding fairness-accuracy curve by varying the fairness constraints is referred to as the **Pareto frontier**. Work including (Kim et al., 2020; Jang et al., 2022) first addresses such topics by solving the MS optimization problem based on a post-processing framework (Hardt et al., 2016) to approximate the MS Pareto frontier. Curves of methods that are located in the upper right region under the frontier correspond to higher values in accuracy and lower values in fairness discrepancy, i.e., a better fairness-accuracy trade-off, as shown in Fig. 2.

Although it provides realistic quantification regarding the fairness-accuracy trade-off, there are several issues with the analysis. First, due to the potential suboptimality of post-processing, the obtained frontier can be suboptimal, with accuracy values along the curve falling below the best achievable. This leads to a troublesome usage of the MS frontier, as the curves of different methods can go beyond the frontier. We refer to Fig. 2 for the demonstration. Second, since the post-processing framework is constructed by random flipping, the feasible regions are empirically determined by the confusion matrix of each sensitive group. This makes post-processing on testing data problematic (Kim et al., 2020; Jang et al., 2022): owing to the discrepancy between training and testing data, the performance on training data and testing data may exhibit variations, leading to infeasible solutions for post-processing on training data, as shown in Fig. 1. Consequently, the obtained frontier does not lead to a meaningful upper bound regarding either the post-processing method or alternative fairness interventions. Recent work characterize the MS Pareto frontier via an in-processing intervention (Dehdashtian et al., 2024). However, as pointed out by the author, the corresponding MS frontier is suboptimal, leading to improper assessment in the Pareto optimal trade-off. Wang et al. (2024) proposes to approximate the model-agnostic (MA) Pareto frontier by directly optimizing the confusion matrix, corresponding to the Bayesian optimal classifier. However, obtaining the Bayesian optimal classifier can be very challenging, and the dependence on the estimation of joint posterior probability can lead to large variations owing to the uncertainty involved. Furthermore, existing work on the Pareto optimality has been limited to the fairness-accuracy trade-off, whereas the Pareto optimal trade-off between fairness notions has not been properly addressed.

In this work, we propose a unified framework to obtain the MS Pareto optimal frontiers regarding both the fairness-accuracy trade-off and the trade-off between fairness notions. We theoretically discuss the suboptimality in (Kim et al., 2020; Jang et al., 2022), where we show the existence of a better MS frontier under mild assumptions. Following the discussions, we reformulate the Pareto optimal trade-offs as constrained optimization problems w.r.t. the *confusion vector* without reliance on specified fairness interventions or exterior estimation tools, bypassing the problem of algorithmic optimality. Based on our reformulation, we propose *fair retraining for the last layer* by adjusting the last linear layer using group-dependent bias. We summarize our contribution as follows:

- We theoretically discuss the suboptimality of the existing MS frontier and propose a reformulation of the MS Pareto frontier.

- We extend our formulation to the trade-off between fairness notions, where we show the dependence of the compatibility of fairness notions on **accuracy**.

- We propose a novel framework for retraining the last layer, and we prove theoretically the superiority of our method over post-processing baselines. Moreover, we show theoretically how our method deviates from the Pareto frontier, validated on real-world datasets.

- We validate from experiments the optimality of our MS Pareto frontier, as well as the superiority of our fair retraining framework in terms of fairness-accuracy trade-off on four benchmark datasets for both binary and multi-class classification tasks.

## 2 RELATED WORK

**Fairness Notions and Trade-offs in Fair Classification**. There are three widely adopted fairness notions in the community: individual fairness (Kusner et al., 2017; Ilvento, 2019; Mukherjee et al., 2020), group fairness (Chouldechova, 2017; Bellamy et al., 2018; Narayanan, 2018), and mini-max fairness (Martinez et al., 2020; Diana et al., 2021; Abernethy et al., 2022; Yang et al., 2023). In this study, we explore group fairness criteria, specifically focused on DP (Dwork et al., 2012) and EOd (Hardt et al., 2016). DP is defined as $\hat{y} \perp\!\!\!\perp a$, where $\hat{y}$ is the prediction of a classifier and $a$ denotes the sensitive attribute, and EOd is defined by $\hat{y} \perp\!\!\!\perp a \mid y$, where $y$ is the ground truth label.

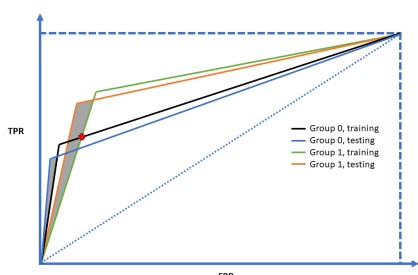

Figure 1: Demonstration of the infeasibility of post-processing on testing data. The feasible regions for post-processing on training and testing data are specified by the triangles of different colors. When the solutions on testing space fall into the non-overlapping parts between training regions and testing regions (the gray regions in the figure), they cannot be achieved by the same post-processing techniques on training data.

Existing work empirically observes that achieving a certain level of fairness typically comes at the cost of accuracy degradation (Calders et al., 2009; Zafar et al., 2017; Menon & Williamson, 2018; Barlas et al., 2021), and achieving certain fairness notion comes at the cost of another fairness notion (Kleinberg et al., 2016; Chouldechova, 2017). In response, Kim et al. (2020); Jang et al. (2022); Dehdashtian et al. (2024); Wang et al. (2024) use the Pareto frontier to characterize this trade-off. Xian et al. (2023) states the inherent trade-off in achieving DP in fair classification. Contrasting with these findings, some studies argue that enforcing fairness in machine learning models does not necessarily reduce utility (Wick et al., 2019; Dutta et al., 2020), and the possibility of improving several fairness notions simultaneously (Reich & Vijaykumar, 2021; Gultchin et al., 2022).

**Post-Processing and Last-Layer Retraining in Fair Classification**. Fairness through post-processing is a prevalent approach in addressing bias in machine learning models. This method involves adjusting the predictions of a pre-trained network to meet specified fairness constraints, typically through random flipping (Hardt et al., 2016) and adjusting thresholds (Corbett-Davies et al., 2017; Menon & Williamson, 2018; Jang et al., 2022; Xian et al., 2023).

Recent work (Menon et al., 2020; Kirichenko et al., 2022) observes that re-training the last linear layer could efficiently correct bias learned by an ERM model. Building on this, LaBonte et al. (2023) suggests that retraining the last linear layer with misclassified data could correct biases inherent in the network. Following this line of research, Chen et al. (2023b) proposes a strategy that leverages machine unlearning, specifically applied to the last linear layer, to effectively eliminate the learned biases. Mao et al. (2023) proposes a last-layer retraining framework incorporating fairness regularizations and reweighting.

| $\alpha_a$ | Base rate of group $a$. |
|---|---|
| $\beta$ | Proportion of samples from sensitive group $a = 1$. |
| $\mathbf{z}$ | Confusion vector. |
| $\mathbf{A}_c$ | Matrix for calculating accuracy from the confusion vector. |
| $\mathbf{A}_{DP}, \mathbf{A}'_{DP}$ | Matrices for calculating DP from the confusion vector. |
| $\mathbf{A}_{EOd}$ | Matrix for calculating EOd from the confusion vector. |
| $\mathbf{z}_b$ | Confusion vector obtained by vanilla training. |
| $\mathbf{z}_b^a$ | Confusion vector obtained by vanilla training on group $a$. |

Table 1: Summarization of notations.

## 3 PROBLEM FORMULATION

Let $\{(x_i, y_i, a_i), 1 \leqslant i \leqslant N\}$ be the training set where $x_i \in \mathbb{R}^d$ is the input feature, $y_i \in \{0, 1\}$ the label and $a_i \in \{0, 1\}$ the sensitive information, let $f_\theta : \mathbb{R}^d \to [0, 1]$ be the function of classifier with $\theta$ the parameters, a fair classification problem can be formulated as

$$\arg \min_\theta \bar{\mathcal{L}}_{cls}, s.t. \bar{\mathcal{L}}_{fair} \leqslant \epsilon, \tag{1}$$

where $\bar{\mathcal{L}}_{cls} = \frac{1}{N} \sum_{i=1}^N \mathcal{L}_{cls}(x_i, y_i; \theta)$ is the average classification loss and $\bar{\mathcal{L}}_{fair} = \frac{1}{N} \sum_{i=1}^N \mathcal{L}_{fair}(x_i, y_i, a_i; \theta)$ is the fairness constraint. Let $\{(x_i^t, y_i^t, a_i^t), 1 \leqslant i \leqslant n\}$ be the testing set, when $\mathcal{L}_{cls}$ is chosen as the classification error and $\mathcal{L}_{fair}$ is chosen as a fairness notion, given a fixed classifier structure $f$ and its associated feasible hypothesis space $\mathcal{H}' \subset \mathcal{H}$ specified by the training data, the following reformulation of equation 1 corresponds to the MS Pareto optimal fairness-accuracy trade-off:

$$\arg \min_{\theta \in \mathcal{H}'} \bar{\mathcal{L}}_{cls}^t, s.t. \bar{\mathcal{L}}_{fair}^t \leqslant \epsilon, \tag{2}$$

where $\bar{\mathcal{L}}_{cls}^t = \frac{1}{n} \sum_{i=1}^n \mathcal{L}_{cls}(x_i^t, y_i^t; \theta)$ and $\bar{\mathcal{L}}_{fair}^t = \frac{1}{n} \sum_{i=1}^n \mathcal{L}_{fair}(x_i^t, y_i^t, a_i^t; \theta)$. We defer the extension to multi-class classification to Appendix 12. The MS trade-off between fairness notions can be similarly formulated based on equation 2. Given the feasible hypothesis space $\mathcal{H}'$ and two fairness notions $\mathcal{L}_{fair}$ and $\mathcal{L}'_{fair}$, we aim to find the best achievable measure w.r.t. one fairness notion under the designated sacrifice in the other fairness notion and in accuracy on the testing set:

$$\arg \min_{\theta \in \mathcal{H}'} \mathcal{L}_{fair}^t, s.t. \mathcal{L}_{fair}^{'t} \leqslant \epsilon, \mathcal{L}_{cls}^t \leqslant \eta. \tag{3}$$

## 4 CHARACTERIZING TRADE-OFFS IN FAIR CLASSIFICATION

### 4.1 PRELIMINARIES

Inspired by (Kim et al., 2020), we observe that accuracy, as well as various notions of fairness, can be represented as linear transformations of the confusion matrix, allowing us to rewrite the intractable problem in equation 2 into simple convex optimization problems. Let $M_a$ be the confusion matrix of group $a$ and $\mathbf{z} = [\text{TPR}_0, \text{TNR}_0, \text{TPR}_1, \text{TNR}_1]^T$ be the corresponding confusion vector by vectorizing $M_a$, Let $\alpha_a := \Pr[Y = 1 | A = a]$ be the base rate of group $a$ and $\beta := \Pr[A = 1]$ be the marginal distribution of sensitive information, we show below reformulations of standard concepts of group fairness and accuracy:

**Accuracy** can be written as Acc $= \mathbf{A}_c \mathbf{z}$, where $\mathbf{A}_c = [\alpha_0(1 - \beta), (1 - \alpha_0)(1 - \beta), \alpha_1 \beta, (1 - \alpha_1)\beta]$ encodes the marginal distribution of data.

**DP** can be expressed as DP $= |\mathbf{A}_{DP}\mathbf{z} + \mathbf{A}'_{DP}(1 - \mathbf{z})|$, where $\mathbf{A}_{DP} = [\alpha_0, 0, -\alpha_1, 0]$ and $\mathbf{A}'_{DP} = [0, (1 - \alpha_0), 0, -(1 - \alpha_1)]$.

**EOd** can be formulated as EOd $= \|\mathbf{A}_{EOd}\mathbf{z}\|_1$, where $\mathbf{A}_{EOd} = \begin{bmatrix} 1 & 0 & -1 & 0 \\ 0 & 1 & 0 & -1 \end{bmatrix}$.

Owing to the intractability of $0 - 1$ loss, it is hard to directly obtain the Pareto optimal trade-off in equation 2. Alternatively, Kim et al. (2020) reduces the Pareto optimal problem in equation 2 to

post-processing a pre-trained classifier on testing data:

$$\arg\min_{\tilde{\mathbf{z}}\in\tilde{\mathcal{K}}}(1 - \mathbf{A}_c\tilde{\mathbf{z}})^2 + \lambda\left\|\mathbf{A}_{\text{EOd}}\tilde{\mathbf{z}}\right\|_2^2, \tag{4}$$

where $\tilde{\mathcal{K}}$ corresponds to the group-dependent parallelograms on the FPR-TPR plane constructed by random flipping (Hardt et al., 2016) and $\lambda$ is the tunable hyperparameter. While such relaxation provides tractable solutions for quantifying the EOd-accuracy trade-off, it fails to quantify the Pareto optimal trade-off between fairness and accuracy, as discussed in the following lemma:

**Assumption 1.** *The group-dependent ROC curves are concave.*

**Lemma 1.** *Under assumption 1, let $\hat{\mathcal{K}}$ be the feasible region constructed by combining random flipping and threshold adjustment, let $\Phi_a : FPR_a(\tau) \rightarrow TPR_a(\tau)$ be the function of ROC curve for group $a$ where $\tau \in [0, 1]$ is the decision threshold, for any $\tilde{\mathbf{z}}$ obtained by equation 4, there always exists a strictly better solution $\hat{\mathbf{z}}$ within $\hat{\mathcal{K}}$:*

$$\mathbf{A}_c\tilde{\mathbf{z}} < \mathbf{A}_c\hat{\mathbf{z}}, \|\mathbf{A}_{EOd}\tilde{\mathbf{z}}\|_1 = \|\mathbf{A}_{EOd}\hat{\mathbf{z}}\|_1, \hat{\mathbf{z}} = \tilde{\mathbf{z}} + [0, \delta, 0, \delta]^T,$$

*where $\delta = \min\{|\Phi_0^{-1}([1, 0, 0, 0]\tilde{\mathbf{z}}) - (1 - [0, 1, 0, 0]\tilde{\mathbf{z}})|, |\Phi_1^{-1}([0, 0, 1, 0]\tilde{\mathbf{z}}) - (1 - [0, 0, 0, 1]\tilde{\mathbf{z}})|\}.$*

We defer full proof to Appendix 14. Regarding Assumption 1, we make this assumption out of the property that the optimal ROC curves of a classifier are expected to be concave (Bradley, 1997). In practice, when the ROC curves are not strictly concave, we still observe a better trade-off under such construction. We refer to Fig. 2 for the empirical validation. Consequently, the MS frontier by equation 4 does not provide an accurate approximation of the MS Pareto optimal fairness-accuracy trade-off, where curves of different fairness interventions can possibly go beyond the frontier, and therefore shall not be thought of as a proper baseline for evaluating the fairness-accuracy trade-offs.

## 4.2 PARETO OPTIMALITY IN FAIRNESS-ACCURACY TRADE-OFF

The MS fairness-accuracy trade-off in equation 2 can be rewritten as $\arg\max_{\theta\in\mathcal{H}'} \mathbf{A}_c\mathbf{z}, s.t. \|\mathbf{A}_{\text{EOd}}\mathbf{z}\|_1 \leqslant \epsilon$, $\arg\max_{\theta\in\mathcal{H}'} \mathbf{A}_c\mathbf{z}, s.t. |\mathbf{A}_{\text{DP}}\mathbf{z} + \mathbf{A}'_{\text{DP}}(1 - \mathbf{z})| \leqslant \epsilon$, where $\mathbf{A}_c$, $\mathbf{A}_{\text{Eod}}$, $\mathbf{A}_{\text{DP}}$ and $\mathbf{A}'_{\text{DP}}$ are determined by the marginal distributions on the testing set, rather than the training set. Owing to the non-linearity between $\mathbf{z}$ and $\theta$, we instead seek to find the supremum of the MS Pareto optimal trade-off. We first state observations that help tighten the upper-bound:

**Optimality of accuracy in vanilla training**. Regarding fairness interventions, imposing non-trivial constraints $\epsilon$ shall result in no better classification loss compared with vanilla training. Consequently, let $\mathbf{z}_b$ be the the confusion vector obtained from the baseline model, we have the following upper-bound regarding $\mathbf{z}$: $\mathbf{A}_c\mathbf{z} \leqslant \mathbf{A}_c\mathbf{z}_b(\hat{\theta})$, $\hat{\theta} = \arg\min_\theta \frac{1}{N}\sum_{i=1}^N \mathcal{L}_{cls}(x_i, y_i)$.

**Optimality of group-dependent accuracy**. The classification accuracy within each sensitive group shall be upper-bounded by the accuracy under vanilla training, and imposing fairness interventions on sensitive groups shall not lead to improved in-group accuracy. Owing to the potential overlap of information between different sensitive groups, we consider two different strategies for upper-bounding group-dependent accuracy, where the supremum is taken over the resulting accuracies: 1. vanilla training on the corresponding group alone; 2. vanilla training on the whole training data. Let $\mathbf{z}_b^a = [\text{TPR}_a, \text{TNR}_a, 0, 0]$ be the confusion vector of group $a$ and $\mathbf{A}_c^a = [\alpha_a, (1 - \alpha_a), 0, 0]$, we have the following upper-bound regarding group-wise accuracy: $\mathbf{A}_c^a\mathbf{z} \leqslant \mathbf{A}_c^a\mathbf{z}_b^a$, $\mathbf{A}_c^a\mathbf{z}_b^a = \max\{\mathbf{A}_c^a\mathbf{z}_b^a(\hat{\theta}), \mathbf{A}_c^a\mathbf{z}_b^a(\hat{\theta}^a)\}$ where $\hat{\theta}^a = \arg\min_\theta \frac{1}{|\{i|a_i=a\}|}\sum_{\{i|a_i=a\}} \mathcal{L}_{cls}(x_i, y_i)$.

Based on the constraints on group-wise accuracy and overall accuracy, we reformulate the optimization problem w.r.t. $\theta$ into the convex optimization problem w.r.t. $\mathbf{z}$ as follows:

$$\arg\max_{\mathbf{z}\in\mathcal{K}} \mathbf{A}_c\mathbf{z}, s.t. \|\mathbf{A}_{\text{EOd}}\mathbf{z}\|_1 \leqslant \epsilon,$$
$$\arg\max_{\mathbf{z}\in\mathcal{K}} \mathbf{A}_c\mathbf{z}, s.t. |\mathbf{A}_{\text{DP}}\mathbf{z} + \mathbf{A}'_{\text{DP}}(1 - \mathbf{z})| \leqslant \epsilon, \tag{5}$$

where $\mathcal{K} := \{\mathbf{z}|0 \leqslant \mathbf{z} \leqslant 1; \mathbf{A}_c\mathbf{z}_b \geqslant \mathbf{A}_c\mathbf{z}; \mathbf{A}_c^a\mathbf{z}_b^a \geqslant \mathbf{A}_c^a\mathbf{z}\}$ specifies the feasible region of $\mathbf{z}$. Since we do not impose any specific fairness relaxations on $f$, our formulation naturally corresponds to the

supremum of the Pareto optimal fairness-accuracy trade-off. The values of $\mathbf{z}_b$ and $\mathbf{z}_b^a$ are determined under multiple randomized initializations of $f$ to approximate the best achievable testing accuracy within $\mathcal{H}'$ specified by training, rather than testing data.

The reformulation in equation 5 provides us with a quick estimation regarding $\mathbf{z}$ due to the low-dimensionality. Specifically, under each inequality constraint $\epsilon_i \in \mathbf{E} := \{\frac{i}{T}|0 \leqslant i \leqslant T\}$ uniformly sampled within the interval $[0, 1]$ where $\frac{1}{T}$ is the sampling interval, we solve the convex optimization problem in equation 5 to approximate the best-achievable accuracy $\mathbf{A}_c\mathbf{z}_i$ under such fairness violation. The corresponding $(\epsilon_i, \mathbf{A}_c\mathbf{z}_i)$'s then form the Pareto optimal frontier on the fairness-accuracy plane.

### 4.3 PARETO OPTIMALITY IN THE TRADE-OFF BETWEEN FAIRNESS NOTIONS

Following our discussion on the fairness-accuracy trade-off, we reformulate the optimization problem in equation 3 w.r.t. $\mathbf{z}$. Specifically, regarding the trade-off between DP and EOd, we characterize the problem as follows:

$$\arg\min_{\mathbf{z} \in \mathcal{K}} |\mathbf{A}_{\text{DP}}\mathbf{z} + \mathbf{A}'_{\text{DP}}(1 - \mathbf{z})| , \ s.t. \ \|\mathbf{A}_{\text{EOd}}\mathbf{z}\|_1 \leqslant \epsilon, \ \mathbf{A}_c\mathbf{z} \geqslant \eta.$$

By varying $\epsilon$ and $\eta$, we are able to approximate the best-achievable trade-off between fairness notions under different accuracy values, corresponding to a set of contours on the EOd-DP plane. Owing to the dependence on $\mathbf{A}_c\mathbf{z}$, one natural question is, under what accuracy does the trade-off between DP and EOd vanish? The following lemma states the incompatibility of achieving both fairness notions simultaneously:

**Lemma 2.** *If $\alpha_0 \neq \alpha_1$, the optimal classifier that achieves both DP and EOd will always have the same accuracy as a constant predictor.*

We defer full proof to Appendix 15. Lemma 2 points to the inherence in the EOd-DP trade-off; consequently, when evaluating fairness interventions, instead of expecting small violations in both DP and EOd, a more reasonable evaluation can be based on the Pareto optimality in the EOd-DP trade-off, where curves of methods that lie closer to their Pareto optimums indicate better improvement in fairness. We refer to Section 6.3 for the detailed discussion.

## 5 FAIR RETRAINING

### 5.1 OPTIMIZATION PROBLEM

Inspired by our formulation of the MS Pareto frontier, we seek to directly optimize over the fairness notions so as to achieve better trade-offs. Specifically, the Lagrangian function of equation 5 can be written as

$$\mathcal{L}_{\text{EOd}}(\mathbf{z}, \lambda) = \mathbf{A}_c\mathbf{z} - (\lambda \|\mathbf{A}_{\text{EOd}}\mathbf{z}\|_1 - \lambda\epsilon),$$
$$\mathcal{L}_{\text{DP}}(\mathbf{z}, \lambda) = \mathbf{A}_c\mathbf{z} - (\lambda |\mathbf{A}_{\text{DP}}\mathbf{z} + \mathbf{A}'_{\text{DP}}(1 - \mathbf{z})| - \lambda\epsilon).$$

For a chosen classifier structure $f$, we may think of it as the composition of two separate models: the encoder $g : \mathbb{R}^d \to \mathbb{R}^m$, which maps the input feature to the latent space, and a linear classification head $h : \mathbb{R}^m \to [0, 1]$, which maps the latent representation to the soft prediction. For a pre-trained fixed encoder $g$, the optimization objective becomes much simpler owing to the linearity of the last layer. Nonetheless, the density estimations in high-dimensional latent space could still be challenging. We therefore consider the following two-layer reformulation of $h$: a linear projection head $h_1 : \mathbb{R}^m \to \mathbb{R}^{\text{dim}}$, which projects the latent representation to a low-dimensional space, and a linear classification head $h_2 : \mathbb{R}^{\text{dim}} \to [0, 1]$. During retraining, we only update the linear classification head $h_2$, while the projection head remains frozen.

Inspired by work on threshold adjustment (Jang et al., 2022), we seek to further increase the degree of freedom when updating the classification head. Let $[\hat{x}_i^0, \hat{x}_i^1] = h_1 \circ g(x_i)$ be the output of $h_1$ and $\hat{x}$ be the input feature of $h_2$, We consider the alternative modelling of the projected data:

$$\hat{x}_i = [\hat{x}_i^0, \hat{x}_i^1, \mathbb{1}[a_i = 0], \mathbb{1}[a_i = 1]].$$

Such formulation allows us to apply group-dependent bias terms, which correlates with tuning group-dependent thresholds. Correspondingly, let $[w, b_0, b_1]$ be the parameters of $h_2$ where $b_a$ represents the

group-dependent bias, let $p_{ya}$ be the estimated distribution of $\hat{x}$ in the subgroup $\{i|y_i = y, a_i = a\}$, we can approximate $\mathbf{z}$ as the integral form: $\text{TPR}_a = \int_{w^T\hat{x}+b_a \geqslant 0} p_{1a}$, $\text{TNR}_a = \int_{w\hat{x}+b_a < 0} p_{0a}$. And the optimization problem regarding $h_2$ can be formulated as

$$\underset{w,b_0,b_1}{\arg\max} \text{Acc}(f_{ya}; w, b_0, b_1) - \lambda\text{EOd}(f_{ya}; w, b_0, b_1), \tag{6}$$

$$\underset{w,b_0,b_1}{\arg\max} \text{Acc}(f_{ya}; w, b_0, b_1) - \lambda\text{DP}(f_{ya}; w, b_0, b_1). \tag{7}$$

Let $\{w^*, b^*\}$ be the parameters of classification head by the baseline model, when the normal vector of $h_2$ is chosen as $w^*$, our formulation is equivalent as thresholding (Jang et al., 2022): $h_2(x_i) = \sigma(w^* x_i + b_a) = \sigma(h_2^*(x_i) + c_a)$, where $c_a = b_a - b^*$ is the group-dependent threshold and $\sigma(x)$ is the sigmoid function.

## 5.2 THEORETICAL ANALYSIS

While our retraining framework naturally includes thresholding, its connection with post-processing under random flipping (Kim et al., 2020) remains unclear. We 'll focus on the EOd optimum, which quantifies the best achievable accuracy under zero EOd:

**Definition 1.** *The EOd optimum $r^*$ of a fairness intervention corresponds to the between-group intersectional point in $FPR - TPR$ plane with the optimal accuracy (e.g., the red star in Fig. 1):*

$$\underset{r^* \in R_f}{\max} Acc, \, s.t. \, EOd = 0,$$

*where $R_f$ is the feasible region of the fairness intervention in $FPR - TPR$ plane determined by the training data.*

And the following theorem states the superiority of our EOd optimum over alternative baselines:

**Theorem 1.** *Under assumption 1, the EOd optimum $\mathbf{z}^*$ by our method achieves strictly better accuracy compared with the EOd optimum $\mathbf{z}'^*$ by random flipping.*

We defer full proof to Appendix 16. Theorem 1 establishes that, under mild assumptions, our method achieves superior performance compared to the average performance of flipping, i.e., random flipping. Specifically, our method attains not only better near-optimality, as proved in (Jang et al., 2022), but also strictly better accuracy in terms of EOd optimum compared with random flipping.

**Comparison with thresholding**. Owing to the group-dependent bias, when choosing $w = w^*$, our method recovers the same ROC curves as the threshold adjustment (Jang et al., 2022), indicating a better or comparable EOd optimum.

We move on to discuss the optimality of our retraining framework compared with the Pareto frontier. Following conventional modelling (Sagawa et al., 2020; Yao et al., 2022; Wang & Wang, 2024), we make the following assumption regarding logits $l$ of each subgroup:

$$l_i \sim \mathcal{N}(\mu_{ya}, s^2), \, i \in \{i|y_i = y, a_i = a\}$$

where the logits of different subgroups share same variance $s^2$ and differ only in group mean $\mu_{ya}$. While the assumption on equal variance seems a bit strong, recent work has pointed to the convergence of group variance under vanilla training (Lu et al., 2024). The following theorem quantifies the accuracy drop of our method where the EOd optimum is attained:

**Lemma 3.** *Let $\gamma := \alpha_0 + \alpha_1\beta - \alpha_0\beta$, under fixed $\mu_{ya}$ and $s$, the group-dependent threshold $c_a^*$ of EOd optimum can be written as*

$$c_0^* = \frac{s^2}{\mu_{11} - \mu_{01}} \log\left(\frac{1-\gamma}{\gamma}\right) + \frac{2\mu_{10} - \mu_{11} + \mu_{01}}{2}, c_1^* = \frac{s^2}{\mu_{11} - \mu_{01}} \log\left(\frac{1-\gamma}{\gamma}\right) + \frac{\mu_{11} + \mu_{01}}{2}.$$

**Theorem 2.** *Let $k = \arg\min_k \int_{-\infty}^{\infty} ||\operatorname{erf}(x) - \tanh(kx)||^2$ and $\mu_y' := k\frac{c_1^* - \mu_{y1}}{\sqrt{2}s}$, let $\mathbf{z}_b$ be the confusion vector under vanilla training and $\mathbf{z}^*$ be the confusion vector of EOd optimum under our method, we have the following estimation of accuracy drop when attaining EOd optimum:*

$$|\mathbf{A}_c\mathbf{z}_b - \mathbf{A}_c\mathbf{z}^*| = \frac{2\mathbf{A}_c\mathbf{z}_b - 1}{2} - \frac{\left(e^{2(\mu_0' + \mu_1')} - 1\right) + (1 - 2\gamma)\left(e^{2\mu_0'} - e^{2\mu_1'}\right)}{\left(e^{2\mu_0'} + 1\right)\left(e^{2\mu_1'} + 1\right)}.$$

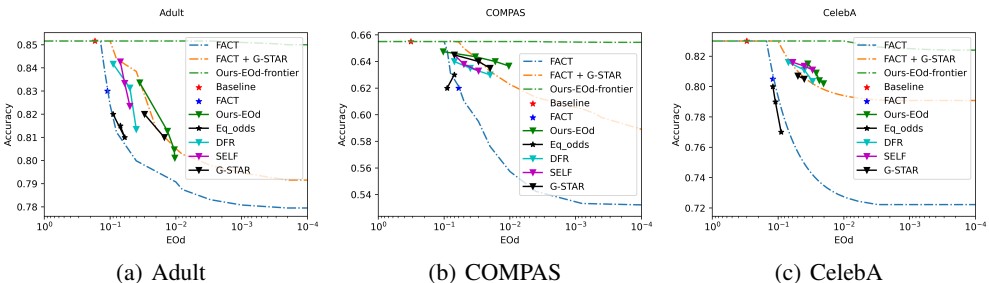

Figure 2: EOd-accuracy trade-off on three datasets. 'Ours-EOd-frontier' corresponds to the MS EOd-accuracy Pareto frontier by equation 5, and 'Ours-EOd' corresponds to our framework in equation 6.

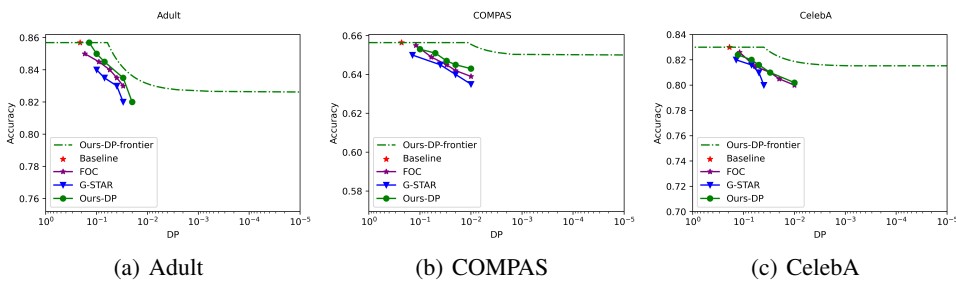

Figure 3: DP-accuracy trade-off on three datasets. 'Ours-DP-frontier' corresponds to the MS DP-accuracy Pareto frontier by equation 5, and 'Ours-DP' corresponds to our framework in equation 7.

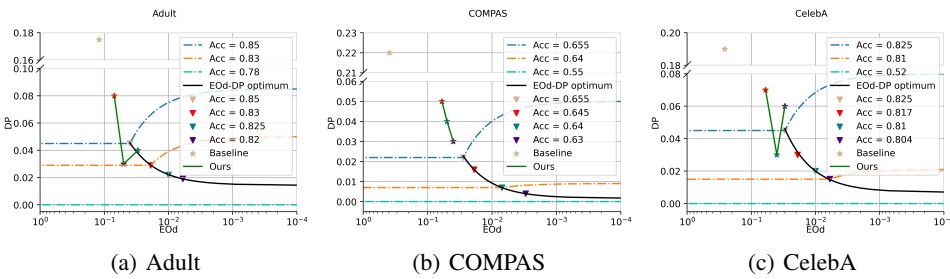

Figure 4: EOd-DP trade-off on three datasets under different accuracy values. The Pareto optimums form a decaying curve on the EOd-DP plane, and the triangles correspond to the Pareto optimums under various accuracy values (tan for the baseline, and the others for our method).

We defer full proof to Appendix 17. Theorem 2 enables us to empirically estimate the decrease in accuracy when achieving the EOd optimum. Specifically, since our method achieves an EOd optimum no worse than thresholding, by estimating $\mu_{ya}$ and $s$ using the latent representation and the classification head $w^*$, we can readily apply Theorem 2 to our retraining framework to obtain the supermum of the accuracy drop on different datasets, without calculating $h_2$. We refer to Appendix 11 for detailed discussion.

## 6 EXPERIMENTS

### 6.1 EXPERIMENTAL SETUP

We validate our method on four benchmark datasets: COMPAS (Larson et al., 2016), Adult (Dua & Graff, 2017), CelebA (Liu et al., 2015) and Drug (Dua & Graff, 2017) [1]. The Drug dataset is for multi-class classification. We use accuracy as utility measure, DP and EOd as fairness measures. We defer details of experimental setup (8), detailed results (9), ablation study (10), empirical validation of theoretical results (11) and the extension to multi-class classification (12) to the Appendix.

We compare our method with the following related methods on post-processing and last-layer retraining: **Baseline**: Neural network without fairness regularization. **FACT**: Post-processing by random flipping (Kim et al., 2020). This method focuses on the EOd-accuracy trade-off. **Eq. Odds**: Post-processing by finding the EOd optimum (Hardt et al., 2016). This method focuses on the EOd-accuracy trade-off. **DFR**: Deep feature reweighting by last-layer retraining (Kirichenko et al., 2022). This method focuses on the EOd-accuracy trade-off and is applied for both **binary and multi-class** classification. **SELF**: Last-layer retraining by selective fine-tuning (LaBonte et al., 2023). This method focuses on the EOd-accuracy trade-off and is applied for both **binary and multi-class** classification. **G-STAR**: Post-processing by thresholding (Jang et al., 2022). This method focuses on both the EOd-accuracy trade-off and the DP-accuracy trade-off. **FOC**: Post-processing based on the score functions (Xian et al., 2023). This method focuses on the DP-accuracy trade-off.

We consider two different approaches in comparison with our MS Pareto frontier for EOd-accuracy trade-off: **FACT**: The objective is optimized over training, rather than testing data. **FACT + G-STAR**: The boundaries of feasible regions are determined by the group-dependent ROC curves, as discussed in Lemma 1. We do not include other frontiers for DP-accuracy trade-off or EOd-DP Trade-off as neither FACT nor G-STAR includes such discussion.

### 6.2 FAIRNESS-ACCURACY TRADE-OFF

Results on EOd-accuracy trade-off are shown in Fig. 2. The objective of our method is chosen as equation 6 in this part. Compared with the frontiers by FACT and by FACT + G-STAR, our MS Pareto frontier shows an almost flat curve, without dramatic decrease in accuracy. Such counter-intuitive results point to the non-inherency of EOd-accuracy trade-off: since the disparities are measured by error rates, instead of the predictions alone, eliminating between-group disparities does not necessarily result in decrease in accuracy. On the contrary, neither FACT nor FACT + G-STAR provides proper approximation of the Pareto optimal EOd-accuracy Trade-off, as curves of several methods go beyond their frontiers. Compared with alternative methods, the curve of our retraining framework lies closer to the MS Pareto frontier, indicating a better EOd-accuracy trade-off. This validates the effectiveness of our group-dependent formulation for bias: compared with last-layer retraining and post-processing, our method obtains better flexibility during the fine-tuning, thereby leading to better or comparable performance in both fairness and accuracy.

Results on DP-accuracy trade-off are shown in Fig. 3. The objective of our method is chosen as equation 7 in this part. Compared with the EOd-accuracy frontier, the DP-accuracy frontier shows a noticeable decrease in accuracy as constraints on DP become more stringent, in line with previous work on the inherent trade-off between DP and accuracy (Zhao & Gordon, 2022). In comparison, our formulation ensures us to quantify the trade-off under dynamic scenarios, rather than under perfect DP only. Compared with other methods on DP, our method shows better or comparable trade-off in terms of both fairness improvement and accuracy, which validates the effectiveness of our method.

### 6.3 EOD-DP TRADE-OFF

Results on EOd-DP trade-off are shown in Fig. 4. When the accuracy is greater than that of a constant predictor, there is a clear trade-off between the two fairness notions, and as the accuracy decreases, the EOd-DP frontiers gradually approach the EOd-axis, indicating that the fairness measures become aligned and no longer present a trade-off when the constraints on accuracy are sufficiently relaxed. Moreover, under each accuracy value, we observe an Pareto optimum (i.e., the inflection point along the curve), where no other points achieves both better DP and better EOd under the same accuracy.

---

[1]Code available at https://github.com/cjy24/Pareto-frontier.

Our observation of such Pareto optimums also indicates a novel quantification for fairness interventions taking into account both fairness notions. Specifically, regarding the EOd-DP curve of certain fairness intervention, we identify its Pareto optimums based on the corresponding accuracy values, and the curve lying closer to its Pareto optimums implies better improvement in fairness. We accordingly modify the objective of our fair retraining framework to take into account both DP and EOd, and compared with the baseline, the curve of our method lies closer to its Pareto optimums, indicating an improvement in fairness.

## 7 CONCLUSION

The trade-off problem in fair classification is important yet less studied. In this paper, we prove theoretically the non-optimality of existing work on the Pareto optimal trade-off, and we propose a novel formulation of MS Pareto optimal trade-off regarding both fairness-accuracy and between fairness notions, which sufficiently approximates the upper-bound of the best achievable trade-off. Based on our formulation, we propose a last-layer retraining framework under group-dependent bias, and we show theoretically the superiority of our retraining framework over post-processing baselines, as well as the deviation of our retraining framework from the Pareto frontier. We validate from experiments that our formulation leads to meaningful and informative quantification of the potential trade-offs for a given classifier, and we show from experiments that our proposed retraining framework achieves better trade-offs compared with state-of-the-art alternatives. Future direction includes model-agnostic Pareto optimal trade-off and extension to the fairness-accuracy trade-off under distribution shift.

## ACKNOWLEDGEMENTS

This work was partially supported by the EMBRIO Institute, contract #2120200, a National Science Foundation (NSF) Biology Integration Institute, NSF IIS #2146091, IIS #2345235, and USDA #2023-67021-41368.

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

## 8 EXPERIMENTAL SUPPLEMENT

### 8.1 DATASETS

**COMPAS**: The COMPAS dataset (Larson et al., 2016) contains 7,215 samples with 11 attributes. Following previous works on fairness (Chouldechova, 2017), we only select black and white defendants in COMPAS dataset, and the modified dataset contains 6,150 samples. The goal is to predict whether a defendant reoffends within two years, and we choose *race* as sensitive attributes.

**Adult**: The Adult dataset (Dua & Graff, 2017) contains 65,123 samples with 14 attributes. The goal is to predict whether an individual's income exceeds $50K$, and we choose *sex* as sensitive attributes.

**CelebA**: The CelebA dataset (Liu et al., 2015) contains 202,599 face images, each of resolution $178 \times 218$, with 40 binary attributes. We choose attractiveness as labels and *sex* as sensitive attributes.

**Drug**: The Drug Consumption dataset (Dua & Graff, 2017) contains 1,885 samples with 12 attributes. The goal is to predict whether an individual has consumed Meth within one year, over one year ago or has never consumed Meth, and we choose *race* as sensitive attributes. This dataset is for multi-class classification.

### 8.2 IMPLEMENTATION

The network structure is chosen as MLP for Adult, COMPAS and Drug datasets and ResNet-50 for CelebA dataset with a $80\%$-$20\%$ training-testing partition of data. We repeat experiments on each dataset three times and report the average results. The input dimension dim of $h_2$ is set as 2 during retraining, and the hyperparameters of compared methods are tuned as suggested by the authors.

We implement our method in PyTorch 2.0.0 on one RTX-3090 GPU. The optimizer for ResNet-50 is chosen as SGD with learning rate $10^{-3}$ and weight decay $10^{-3}$. For MLP, the optimizer is chosen as SGD with learning rate $10^{-2}$ and momentum 0.8. The hyperparameter range for our method is set as $(0, 10)$.

## 9 AUXILIARY EXPERIMENTAL RESULTS

We include the detailed results in terms of EOd, DP and accuracy in Tab. 2 - 7.

| Method | Accuracy | EOd |
|---|---|---|
| Baseline | 65.58±1.21% | 38.89±2.72% |
| FACT (Kim et al., 2020) | 62.35±0.65% | 7.34±1.15% |
| Eq. Odds (Hardt et al., 2016) | 62.34±0.56% | 9.92±1.14% |
| DFR (Kirichenko et al., 2022) | 63.73±0.39% | 3.34±1.52% |
| SELF (LaBonte et al., 2023) | 63.68±0.82% | 4.95±1.29% |
| G-STAR (Jang et al., 2022) | 63.62±0.46% | 2.18±0.55% |
| Ours | **63.91±0.53%** | **1.04±0.52%** |

Table 2: Experimental results of EOd-accuracy trade-off on COMPAS dataset.

| Method | Accuracy | EOd |
|---|---|---|
| Baseline | 85.16±0.30% | 20.72±1.81% |
| FACT (Kim et al., 2020) | 82.87±0.86% | 10.29±1.56% |
| Eq. Odds (Hardt et al., 2016) | 82.12±0.77% | 10.75±1.56% |
| DFR (Kirichenko et al., 2022) | 83.18±1.28% | 5.44±1.28% |
| SELF (LaBonte et al., 2023) | 83.51±1.51% | 7.14±0.91% |
| G-STAR (Jang et al., 2022) | 82.13±1.25% | **2.95±0.75%** |
| Ours | **83.47±0.53%** | 3.11±0.54% |

Table 3: Experimental results of EOd-accuracy trade-off on Adult dataset.

| Method | Accuracy | EOd |
|---|---|---|
| Baseline | 82.67±0.63% | 26.42±1.61% |
| FACT (Kim et al., 2020) | 80.87±1.18% | 12.29±1.12% |
| Eq. Odds (Hardt et al., 2016) | 79.56±0.59% | 11.76±1.44% |
| DFR (Kirichenko et al., 2022) | 81.24±0.81% | 4.79±1.67% |
| SELF (LaBonte et al., 2023) | **81.68±1.26%** | 4.14±0.85% |
| G-STAR (Jang et al., 2022) | 80.84±1.27% | 5.45±0.86% |
| Ours | 81.53±0.77% | **3.04±0.85%** |

Table 4: Experimental results of EOd-accuracy trade-off on CelebA dataset.

| Method | Accuracy | DP |
|---|---|---|
| Baseline | 65.58±1.21% | 22.89±1.54% |
| FOC (Xian et al., 2023) | 63.96±0.75% | **1.06±0.55%** |
| G-STAR (Jang et al., 2022) | 64.08±0.62% | 2.31±0.32% |
| Ours | **64.24±0.64%** | 1.31±0.46% |

Table 5: Experimental results of DP-accuracy trade-off on COMPAS dataset.

| Method | Accuracy | DP |
|---|---|---|
| Baseline | 85.16±0.30% | 17.67±1.44% |
| FOC (Xian et al., 2023) | 83.25±0.69% | 3.17±1.17% |
| G-STAR (Jang et al., 2022) | 82.20±0.44% | 3.16±0.93% |
| Ours | **83.64±0.72%** | **3.03±0.76%** |

Table 6: Experimental results of DP-accuracy trade-off on Adult dataset.

| Method | Accuracy | DP |
|---|---|---|
| Baseline | 82.67±0.63% | 18.62±1.34% |
| FOC (Xian et al., 2023) | **81.27±0.36%** | 3.14±0.65% |
| G-STAR (Jang et al., 2022) | 81.13±0.39% | 5.16±0.84% |
| Ours | 81.17±0.55% | **2.93±0.53%** |

Table 7: Experimental results of DP-accuracy trade-off on CelebA dataset.

## 10 ABLATION STUDY

Regarding our retraining framework, the input dimension of $h_2$ can vary based on the linear projection head $h_1$. Results of varying the input dimension are shown in Tab. 8-9. Compared with other choices (dim = 4, dim = 6), dim = 2 in general provides better or comparable performance in terms of fairness and accuracy, which validates the feasibility of our framework.

| Method | Accuracy | DP | EOd |
|---|---|---|---|
| Ours (dim=2) | 64.67±0.77% | 3.31±0.59% | 1.04±0.52% |
| Ours (dim=4) | 64.24±0.40% | 3.79±0.62% | 1.59±0.49% |
| Ours (dim=6) | 64.72±0.63% | 4.45±0.67% | 1.33±0.56% |

Table 8: Experimental results on COMPAS dataset. The distribution of each subgroup is approximated by multivariate normal distribution.

| Dataset | Method | Accuracy | DP | EOd |
|---|---|---|---|---|
| Adult | Baseline | 85.16±0.30% | 17.67±1.44% | 20.72±1.81% |
| Adult | Ours (dim=2) | 83.36±0.51% | 4.43±1.28% | 3.54±1.59% |
| Adult | Ours (dim=4) | 83.40±0.31% | 5.28±1.78% | 4.41±1.44% |
| Adult | Ours (dim=6) | 83.16±0.83% | 5.32±1.36% | 6.23±1.86% |

Table 9: Experimental results on Adult dataset. The distribution of each subgroup is approximated by multivariate normal distribution.

Moreover, it is possible to use multiple classification heads $h_2$ for different groups. The optimization problem then becomes

$$\arg\max_{w_0,w_1,b_0,b_1} \mathrm{Acc}(f_{ya}; w_0, w_1, b_0, b_1) - \lambda \mathrm{EOd}(f_{ya}; w_0, w_1, b_0, b_1),$$

Results of using multiple classification heads are shown in Tab. 10. Compared with using single classification head, using group-dependent classification head achieves marginal improvement in fairness.

| Dataset | Method | Accuracy | DP | EOd |
|---------|--------|----------|-----|-----|
| COMPAS | Baseline | 65.58±1.21% | 22.89±1.54% | 38.89±2.72% |
| COMPAS | Ours ($w$) | 64.67±0.77% | 3.31±0.59% | 1.04±0.52% |
| COMPAS | Ours ($w_a$'s) | 64.74±0.64% | 3.24±0.32% | 1.02±0.57% |
| Adult | Baseline | 85.16±0.30% | 17.67±1.44% | 20.72±1.81% |
| Adult | Ours ($w$) | 83.36±0.51% | 4.43±1.28% | 3.54±1.06% |
| Adult | Ours ($w_a$'s) | 83.47±0.53% | 4.12±1.46% | 3.11±0.54% |

Table 10: Experimental results on COMPAS and Adult dataset. The distribution of each subgroup is approximated by 2D Gaussian. Experiments are repeated three times.

## 11 EMPIRICAL VERIFICATION OF THEORETICAL RESULTS

We estimate the accuracy drop relative to vanilla training when attaining the EOd optimum using our method. Following the discussion in Theorem 2, we leverage the pre-trained encoder and linear classifier $w^*$ to estimate the values of $\mu'_y$. Results are shown in Tab. 11. Compared to the results on the COMPAS and CelebA datasets, our method on the Adult dataset exhibits a larger accuracy drop when attaining the EOd optimum under retraining, consistent with our empirical observation that fair retraining on Adult dataset leads to larger deviation from the Pareto frontier compared with the other two datasets.

| Dataset | $\mathbf{A}_c\mathbf{z}^*$ | $\left|\mathbf{A}_c\mathbf{z}_b - \mathbf{A}_c\mathbf{z}^*\right|$ |
|---------|------|------|
| COMPAS | 64.5% | 1.3% |
| CelebA | 79.73% | 2.94% |
| Adult | 80.87% | 4.36% |

Table 11: Estimations of accuracy drop when achieving EOd optimums on three datasets. $\mathbf{A}_c\mathbf{z}^*$ is estimated by $\omega(\mu'_0, \mu'_1) + \frac{1}{2}$ in Theorem 2.

## 12 EXTENSION TO MULTI-CLASS CLASSIFICATION

We move on to discuss the extension to non-binary classification. Given $y \in [k]$ the class label, let $\text{Acc}_{ya} := \Pr[f(x) = y|y, a]$ be the classification accuracy in the subgroup $\{i|y_i = y, a_i = a\}$ and $\mathbf{z} := [\text{Acc}_{10}, \ldots, \text{Acc}_{k0}, \text{Acc}_{11}, \ldots, \text{Acc}_{k1}]$ the corresponding confusion vector, let $\{\alpha_{1a}, \ldots, \alpha_{ka}\}$ be the marginal distribution of label in group $a$, the DP and EOd constraints under multi-class classification (Denis et al., 2021) can be formulated as

$$\text{EOd}: \max_{k' \in [k]} |\mathbf{A}_{\text{EOd},k'}\mathbf{z}| \leqslant \epsilon,$$
$$\text{DP}: \max_{k' \in [k]} |\mathbf{A}_{\text{DP},k'}\mathbf{z} - \mathbf{A}'_{\text{DP},k'}(1 - \mathbf{z})| \leqslant \epsilon,$$
(8)

where $\mathbf{A}_{\text{EOd},k'} = [0, \ldots, \mathbb{1}[y = k', a = 0], \ldots, \mathbb{1}[y = k', a = 1], \ldots, 0]$, $\mathbf{A}_{\text{DP},k'} = [0, \ldots, \alpha_{k'0}\mathbb{1}[y = k', a = 0], \ldots, \alpha_{k'1}\mathbb{1}[y = k', a = 1], \ldots, 0]$ and $\mathbf{A}'_{\text{DP},k'} = \mathbb{1}[y \neq k'][\alpha_{10}, \ldots, \alpha_{k0}, \alpha_{11}, \ldots, \alpha_{k1}]$. Regarding multi-class classification, the disparities are quantified by the largest discrepancy amongst all the subgroups. Accordingly, the MS Pareto optimal fairness-accuracy trade-off can be formulated as

$$\arg\max_{\mathbf{z} \in \mathcal{K}} \mathbf{A}_c\mathbf{z}, \ s.t. \ |\mathbf{A}_{\text{EOd},k'}\mathbf{z}|_1 \leqslant \epsilon, \forall k' \in [k],$$

$$\arg\max_{\mathbf{z} \in \mathcal{K}} \mathbf{A}_c\mathbf{z}, \ s.t. \ \left|\mathbf{A}_{\text{DP},k'}\mathbf{z} + \mathbf{A}'_{\text{DP},k'}(1 - \mathbf{z})\right| \leqslant \epsilon, \forall k' \in [k],$$

where $\mathbf{A}_c = [(1-\beta)\alpha_{10}, \ldots, (1-\beta)\alpha_{k0}, \beta\alpha_{11}, \ldots, \beta\alpha_{k1}]$, and the fairness constraints are taken over all the subgroups to accommodate with the maximum notions in equation 8. Let $b_a = [b_{1a}, \ldots, b_{ka}]$ be the bias vector for group $a$, we similarly formulate our fair retraining framework as follows:

$$\arg\max_{w, b_0, b_1} \text{Acc}(f_{ya}; w, b_0, b_1) - \lambda \text{EOd}(f_{ya}; w, b_0, b_1),$$

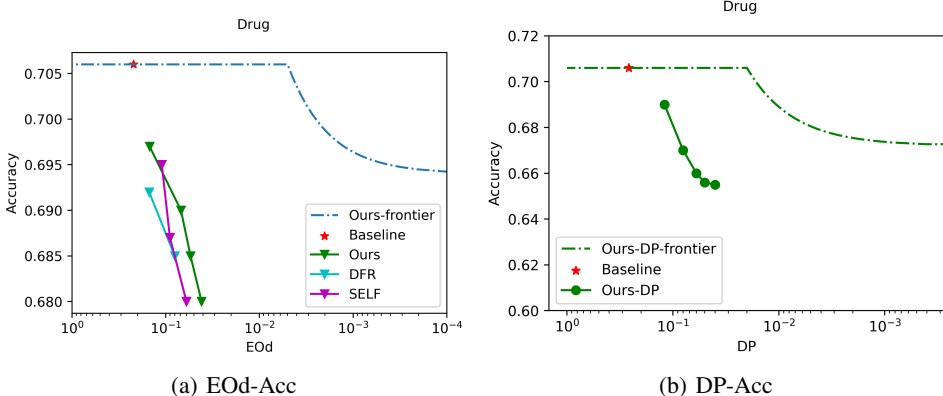

(a) EOd-Acc               (b) DP-Acc

Figure 5: EOd-accuracy trade-off and DP-accuracy trade-off on Drug dataset. We extend our discussion to multi-class setting compared with existing works on fairness-accuracy trade-off.

$$\arg\max_{w,b_0,b_1} \mathrm{Acc}(f_{ya}; w, b_0, b_1) - \lambda \mathrm{DP}(f_{ya}; w, b_0, b_1),$$

Regarding multi-class classification, we do not consider the post-processing baselines for comparison as neither includes a direct extension to such tasks. We consider two benchmarks:

- **DFR**: Deep feature reweighting by last-layer retraining (Kirichenko et al., 2022).
- **SELF**: Last-layer retraining by selective fine-tuning (LaBonte et al., 2023).

Results on EOd-accuracy trade-off and DP-accuracy trade-off for multi-class classification are shown in Fig. 5. Compared with alternative methods for last-layer retraining, our method achieves better EOd-accuracy trade-off. Moreover, compared with the baseline, our method shows improvement in DP with relatively small degradation in accuracy. This validates the effectiveness of our method under multi-class classification. Detailed results are in Tab. 12-13.

| Method | Accuracy | EOd |
|---|---|---|
| Baseline | 70.63±0.79% | 22.57±1.96% |
| DFR (Kirichenko et al., 2022) | 67.64±0.55% | 6.17±1.11% |
| SELF (LaBonte et al., 2023) | 68.13±0.58% | 6.34±0.87% |
| Ours | **68.76±0.73%** | **5.04±0.95%** |

Table 12: Experimental results of EOd-accuracy trade-off on Drug dataset.

| Method | Accuracy | DP |
|---|---|---|
| Baseline | 70.63±0.79% | 26.34±2.25% |
| Ours | 66.48±0.67% | **5.67±0.78%** |

Table 13: Experimental results of DP-accuracy trade-off on Drug dataset.

## 13 EXTENSION TO ALTERNATIVE NOTIONS

Our formulation can be readily generalized to alternative group fairness notions. For example, the calibration-within-groups notion (Kim et al., 2020) can be formulated as

$$\mathrm{CG} = ||\mathbf{A}_{\mathrm{CG}}(\mathbf{A}_0 \mathbf{z}_1 + \mathbf{A}_1(1 - \mathbf{z}_1))||_1,$$

where

$$\mathbf{A}_{\mathrm{CG}} = \begin{bmatrix} 1-v_1 & 0 & -v_1 & 0 & 0 & 0 & 0 & 0 \\ 0 & 1-v_0 & 0 & -v_0 & 0 & 0 & 0 & 0 \\ 0 & 0 & 0 & 0 & 1-v_1 & 0 & -v_1 & 0 \\ 0 & 0 & 0 & 0 & 0 & 1-v_0 & 0 & -v_0 \end{bmatrix},$$

$$\mathbf{A}_0 = \begin{bmatrix} \alpha_0(1-\beta) & 0 & 0 & 0 \\ 0 & 0 & 0 & 0 \\ 0 & 0 & 0 & 0 \\ 0 & (1-\alpha_0)(1-\beta) & 0 & 0 \\ 0 & 0 & \alpha_1\beta & 0 \\ 0 & 0 & 0 & 0 \\ 0 & 0 & 0 & 0 \\ 0 & 0 & 0 & (1-\alpha_1)\beta \end{bmatrix},$$

$$\mathbf{A}_1 = \begin{bmatrix} 0 & 0 & 0 & 0 \\ \alpha_0(1-\beta) & 0 & 0 & 0 \\ 0 & (1-\alpha_0)(1-\beta) & 0 & 0 \\ 0 & 0 & 0 & 0 \\ 0 & 0 & 0 & 0 \\ 0 & 0 & \alpha_1\beta & 0 \\ 0 & 0 & 0 & (1-\alpha_1)\beta \\ 0 & 0 & 0 & 0 \end{bmatrix}.$$

Specifically, $\alpha_a = \Pr[Y = 1 | A = a]$ and $\beta = \Pr[A = 1]$, and $v_0$, $v_1$ are score representatives such that $0 \leqslant v_0, v_1 \leqslant 1$.

## 14 PROOF FOR LEMMA 1

*Proof.* Let $\hat{y}$ be the original prediction and $\tilde{y}$ be the flipped prediction, let $\widetilde{\mathrm{mea}}_a$ be the measures under flipping and $\mathrm{mea}_a$ be the measures without flipping, we have

$$P[\tilde{y} = 1 | y, a] = P[\tilde{y} = 1 | \hat{y} = 1, a]P[\hat{y} = 1 | y, a] + P[\tilde{y} = 1 | \hat{y} = 0, a]P[\hat{y} = 0 | y, a].$$

Therefore, for $y = 1$, we have

$$\widetilde{\mathrm{TPR}}_a = P[\tilde{y} = 1 | y = 1, a] = P[\tilde{y} = 1 | \hat{y} = 1, a]\mathrm{TPR}_a + P[\tilde{y} = 1 | \hat{y} = 0, a]\mathrm{FNR}_a.$$

And for $y = 0$, we have

$$\widetilde{\mathrm{FPR}}_a = P[\tilde{y} = 1 | y = 0, a] = P[\tilde{y} = 1 | \hat{y} = 1, a]\mathrm{FPR}_a + P[\tilde{y} = 1 | \hat{y} = 0, a]\mathrm{TNR}_a.$$

Consequently, the feasible region of (Kim et al., 2020) is determined by the group-dependent parallelograms in the FPR − TPR plane with the vertices as

$$\{(0, 0), (\mathrm{FPR}_0, \mathrm{TPR}_0), (\mathrm{TNR}_0, \mathrm{FNR}_0), (1, 1)\}, \text{for group 0},$$

$$\{(0, 0), (\mathrm{FPR}_1, \mathrm{TPR}_1), (\mathrm{TNR}_1, \mathrm{FNR}_1), (1, 1)\}, \text{for group 1},$$

where $\{\mathrm{FPR}_a, \mathrm{TPR}_a, \mathrm{TNR}_a, \mathrm{FNR}_a\}$ are obtained from the baseline model. We first make a few simplifications regarding the feasible regions of the parallelograms.

**Proposition 1.** *The $(\widetilde{\mathrm{FPR}}_a, \widetilde{\mathrm{TPR}}_a)$ solution pair under random flipping always fall into the upper-half of the parallelograms.*

For any $\{r_0, r_1\} := \{(\widetilde{\mathrm{FPR}}_a, \widetilde{\mathrm{TPR}}_a), 0 \leqslant a \leqslant 1\}$ pair in the lower-half of the parallelograms, since the solutions lie below the line segment TPR = FPR, we have

$$\widetilde{\mathrm{FPR}}_a > \widetilde{\mathrm{TPR}}_a, \forall a. \tag{9}$$

Consider the symmetric point $r_a^s$ of $r_a$ about the line segment TPR = FPR, we have $r_a^s = (\widetilde{\mathrm{TPR}}_a, \widetilde{\mathrm{FPR}}_a)$. Consequently, by equation 9 we have

$$\mathrm{EOd}(r_a^s) = |\widetilde{\mathrm{TPR}}_0 - \widetilde{\mathrm{TPR}}_1| + |\widetilde{\mathrm{FPR}}_0 - \widetilde{\mathrm{FPR}}_1| = \mathrm{EOd}(r_a),$$

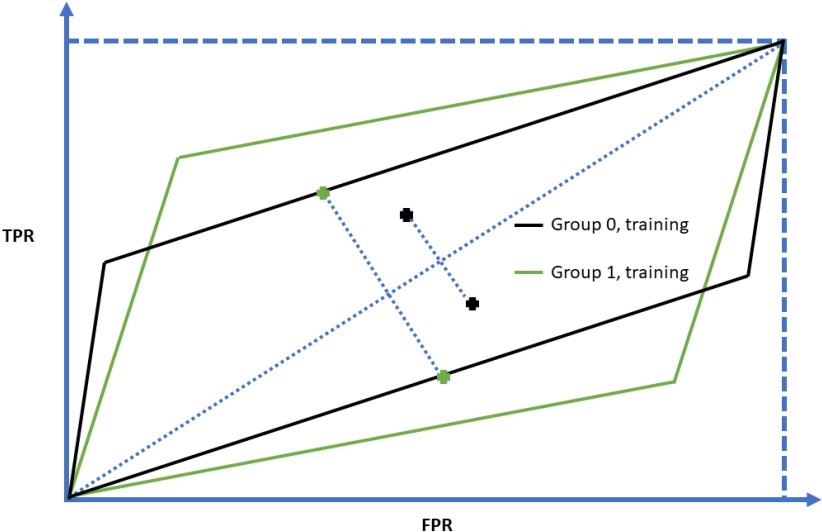

Figure 6: Demonstration of the $(\tilde{\text{FPR}}_a, \tilde{\text{TPR}}_a)$ solution. The solution pair in the upper-half regions indicates better accuracy (higher TPR and lower FPR) and same EOd owing to the symmetry.

$$\text{Acc}(r_a^s) = (1 - \beta)(\alpha_0\tilde{\text{FPR}}_0 + (1 - \alpha_0)(1 - \tilde{\text{TPR}}_0)) + \beta(\alpha_1\tilde{\text{FPR}}_1 + (1 - \alpha_1)(1 - \tilde{\text{TPR}}_1))$$
$$> (1 - \beta)(\alpha_0\tilde{\text{TPR}}_0 + (1 - \alpha_0)(1 - \tilde{\text{FPR}}_0)) + \beta(\alpha_1\tilde{\text{TPR}}_1 + (1 - \alpha_1)(1 - \tilde{\text{FPR}}_1)) = \text{Acc}(r_a).$$

This shows the existence of strictly better solutions in the upper-half of the parallelograms under the symmetric construction, leading to the same EOd but better accuracy, as demonstrated in Fig. 6.

**Proposition 2.** *At least one of the $(\tilde{\text{FPR}}_a, \tilde{\text{TPR}}_a)$ solution fall on the boundary of the parallelograms.*

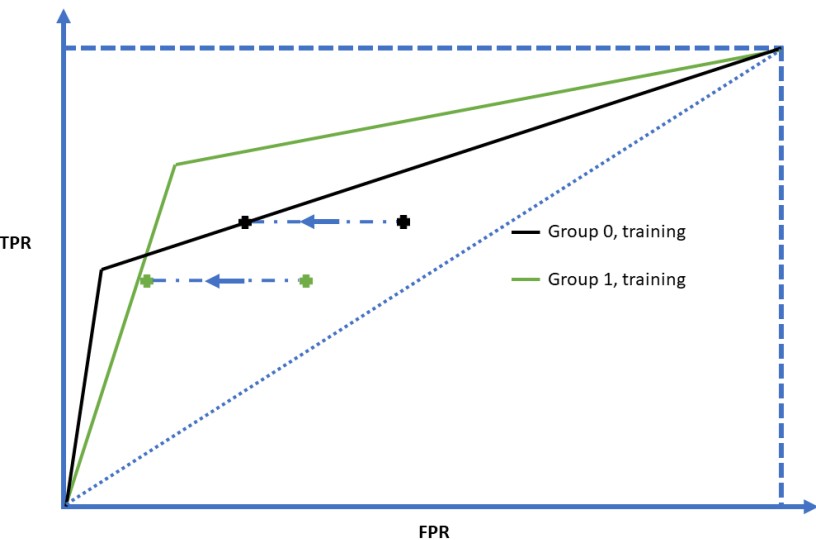

Figure 7: Demonstration of translation along the FPR-axis. The solution pair after translation obtains better accuracy (same TPR and lower FPR) and same EOd owing to the equidistant translations.

For any $\{r_0, r_1\} := \{(\tilde{\text{FPR}}_a, \tilde{\text{TPR}}_a), 0 \leqslant a \leqslant 1\}$ pair where both $r_0$ and $r_1$ lie within the group-dependent parallelograms, we consider the following equidistant translation along the FPR-axis:

$$r'_a = (\tilde{\text{FPR}}_a - \xi, \tilde{\text{TPR}}_a), \ \xi = \min_a \left\{\tilde{\text{FPR}}_a - \frac{\text{FPR}_a}{\text{TPR}_a}\tilde{\text{TPR}}_a, \tilde{\text{FPR}}_a - \frac{1 - \text{FPR}_a}{1 - \text{TPR}_a}(\tilde{\text{TPR}}_a - \text{TPR}_a) - \text{FPR}_a\right\}.$$

Under such translation, at least one of the solution $r'_a$ touches the boundary of its parallelogram. This leads to the same EOd but better accuracy, owing to the decrease in FPR. Therefore, any solution pair that lie within the group-dependent parallelograms will be sub-optimal, as shown in Fig. 7.

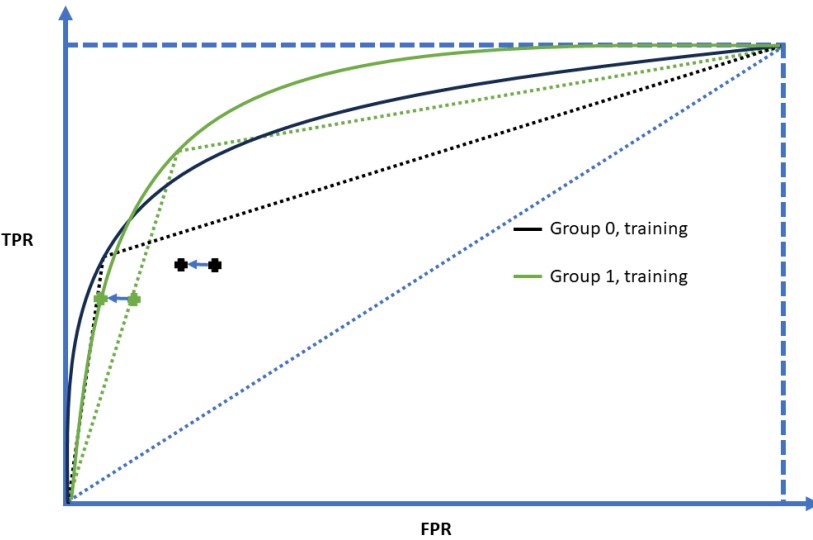

Figure 8: Considering the combination of random flipping and threshold adjustment, for any solution pair obtained by random flipping, there always exist a strictly better solution under equidistant translation along FPR-axis, where at least one of the new solution points touches the group-dependent ROC curve.

By Prop. 1 and 2, we may simplify the discussion to the boundary of different fairness interventions. Since the ROC curve of each sensitive group is concave, let $\text{ROC}_a$ be the ROC curve of group $a$, consider the combination of group-dependent thresholding and random flipping, the feasible region for each sensitive group now becomes a convex hull with its boundary the ROC curves, rather than line segments. Moreover, since both the ROC curves and the group-dependent parallelograms pass $\{(\text{FPR}_a, \text{TPR}_a), (\text{TNR}_a, \text{FNR}_a)\}$, owing to the concavity of ROC curves, for any solution pair $(\tilde{\text{FPR}}_a, \tilde{\text{TPR}}_a)$ obtained from FACT (Kim et al., 2020), we can always translate them along the direction of the negative half of the FPR axis up to distance $\delta$ such that at least one of the solution touches the ROC curve, as shown in Fig. 8:

$$\delta := \min\{|\Phi_0^{-1}(\tilde{\text{TPR}}_0) - \tilde{\text{FPR}}_0|, |\Phi_1^{-1}(\tilde{\text{TPR}}_1) - \tilde{\text{FPR}}_1|\}.$$

Consequently, regarding EOd, we have

$$|\tilde{\text{TPR}}_0 - \tilde{\text{TPR}}_1| + |\tilde{\text{FPR}}_0 - \tilde{\text{FPR}}_1| = |\tilde{\text{TPR}}_0 - \tilde{\text{TPR}}_1| + |(\tilde{\text{FPR}}_0 - \delta) - (\tilde{\text{FPR}}_1 - \delta)|.$$

And regarding the accuracy, we have

$$(1 - \beta)(\alpha_0\tilde{\text{TPR}}_0 + (1 - \alpha_0)(1 - \tilde{\text{FPR}}_0)) + \beta(\alpha_1\tilde{\text{TPR}}_1 + (1 - \alpha_1)(1 - \tilde{\text{FPR}}_1))$$
$$< (1 - \beta)(\alpha_0\tilde{\text{TPR}}_0 + (1 - \alpha_0)(1 - \tilde{\text{FPR}}_0 + \delta)) + \beta(\alpha_1\tilde{\text{TPR}}_1 + (1 - \alpha_1)(1 - \tilde{\text{FPR}}_1 + \delta)).$$

Therefore, such translation preserves the EOd gap but improves accuracy, thereby indicating a strictly better EOd-accuracy trade-off than (Kim et al., 2020). $\quad\square$

## 15 PROOF FOR LEMMA 2

*Proof.* Consider the confusion vector $\mathbf{z} = [\text{TPR}_0, \text{TNR}_0, \text{TPR}_1, \text{TNR}_1]^T$. Under zero violation in EOd, we have

$$\text{TPR}_0 = \text{TPR}_1 = \text{TPR}, \ \text{TNR}_0 = \text{TNR}_1 = \text{TNR}.$$

Correspondingly, let $\alpha_a := \Pr[Y = 1 | A = a]$ be the base rate of group $a$, under zero violation in DP, we have

$$\alpha_0 \text{TPR}_0 + (1 - \alpha_0)(1 - \text{TNR}_0) = \alpha_1 \text{TPR}_1 + (1 - \alpha_1)(1 - \text{TNR}_1). \tag{10}$$

Since $\alpha_0 \neq \alpha_1$, we can further simplify equation 10 as

$$\text{TPR} + \text{TNR} = 1.$$

Let $\alpha := \Pr[Y = 1]$ be the base rate of testing data, we have the accuracy under zero violation of DP and EOd as

$$\text{Acc} = \alpha \text{TPR} + (1 - \alpha)(1 - \text{TPR}) = (2\alpha - 1)\text{TPR} + (1 - \alpha).$$

This shows that under zero violation of DP and EOd, the accuracy can be written as a linear function w.r.t. TPR. When $\alpha \geqslant 0.5$, the best accuracy will be $\alpha$, which corresponds to an all-positive predictor. When $\alpha < 0.5$, the best accuracy will be $(1 - \alpha)$, corresponding to an all-negative predictor. $\square$

## 16 PROOF FOR THEOREM 1

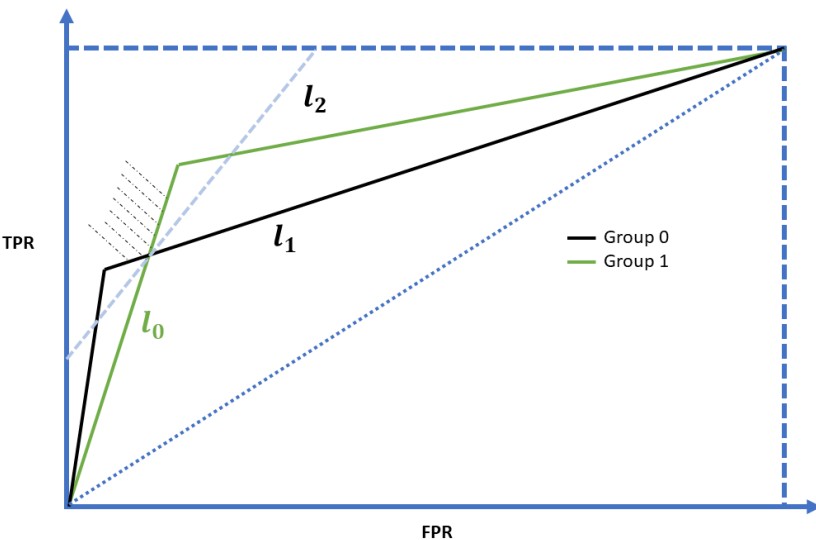

Figure 9: Demonstration of the feasible regions by random flipping. The line segment $l_0$ passes through $(0, 0)$ and $(\text{FPR}_0, \text{TPR}_0)$, and line segment $l_1$ passes through $(\text{FPR}_1, \text{TPR}_1)$ and $(1, 1)$.

*Proof.* Let $\alpha := \Pr[y = 1]$ be the base rate of training data, the EOd optimums under different accuracy values can be expressed as line segments in the $\text{FPR} - \text{TPR}$ plane:

$$\text{TPR} = \frac{1 - \alpha}{\alpha} \text{FPR} + \frac{\text{Acc} - (1 - \alpha)}{\alpha}.$$

We denote such line segment passing through the EOd optimum by random flipping as $l_2$. When $\alpha \leqslant \frac{1}{2}$, so long as the EOd optimum by random flipping achieves better accuracy than constant prediction, i.e., $\text{Acc} \geqslant (1 - \alpha)$, the intercept of $l_2$ will be strictly positive, and the slope of $l_2$ is strictly smaller than $l_0$. When $\alpha > \frac{1}{2}$, when setting $\text{FPR} = 1$, we have $\text{TPR} = \frac{\text{Acc}}{\alpha} > 1$, which indicates that the slope of $l_2$ shall be strictly greater than $l_0$ as shown in Fig. 9. Consequently, $l_2$ always fall within the region between $l_0$ and $l_1$, where the slopes of line segments passing through the

EOd optimum by random flipping are smaller than $l_0$ but greater than $l_1$. Regarding the EOd optimum by threshold adjustment, since it always fall within the upper-left region of the EOd optimum by random flipping, i.e., the shady region in Fig. 9, as proved in Theorem 5 of (Jang et al., 2022), it always lead to better accuracy compared with the EOd optimum by random flipping as $\alpha$ remains constant. Moreover, as the EOd optimum achieves no better accuracy than our method, we thereby conclude that the EOd optimum $r^*$ by our method achieves strictly better accuracy compared with $r'^*$ by random flipping. $\qquad\square$

## 17 PROOF FOR LEMMA 3 AND THEOREM 2

*Proof.* Given group mean $\mu_{ya}$ and variance $s^2$, under group-dependent thresholding, we have the group-wise accuracy as

$$\widetilde{\mathrm{Acc}}_{ya} = \begin{cases} \dfrac{1}{2} + \dfrac{1}{2}\mathrm{erf}\left(-\dfrac{c_a - \mu_{ya}}{\sqrt{2}s}\right), \text{ for } y = 0 \\ \dfrac{1}{2} - \dfrac{1}{2}\mathrm{erf}\left(-\dfrac{c_a - \mu_{ya}}{\sqrt{2}s}\right), \text{ for } y = 1 \end{cases}$$

Since $s$ remains constant before and after thresholding, to achieve zero EOd, we have

$$c_0 - \mu_{10} = c_1 - \mu_{11},$$
$$c_0 - \mu_{00} = c_1 - \mu_{01},$$

which simplifies to $\mu_{10} - \mu_{11} = \mu_{00} - \mu_{01}$. Let $\alpha_a := P[y = 1 \mid A = a]$ be the base rate within group $a$ and let $\beta := P[A = 1]$, we have the accuracy under post-processing as

$$\widetilde{\mathrm{Acc}} = \alpha_1\beta\widetilde{\mathrm{Acc}}_{11} + (1-\alpha_1)\beta\widetilde{\mathrm{Acc}}_{01} + \alpha_0(1-\beta)\widetilde{\mathrm{Acc}}_{10} + (1-\alpha_0)(1-\beta)\widetilde{\mathrm{Acc}}_{00}$$

$$= \alpha_1\beta\left(\frac{1}{2} - \frac{1}{2}\mathrm{erf}\left(-\frac{c_1 - \mu_{11}}{\sqrt{2}s}\right)\right) + (1-\alpha_1)\beta\left(\frac{1}{2} - \frac{1}{2}\mathrm{erf}\left(-\frac{c_1 - \mu_{01}}{\sqrt{2}s}\right)\right)$$

$$+ \alpha_0(1-\beta)\left(\frac{1}{2} - \frac{1}{2}\mathrm{erf}\left(-\frac{c_1 - \mu_{10}}{\sqrt{2}s}\right)\right) + (1-\alpha_0)(1-\beta)\left(\frac{1}{2} - \frac{1}{2}\mathrm{erf}\left(-\frac{c_1 - \mu_{00}}{\sqrt{2}s}\right)\right).$$

Since $\widetilde{\mathrm{Acc}}_{11} = \widetilde{\mathrm{Acc}}_{10}$ and $\widetilde{\mathrm{Acc}}_{01} = \widetilde{\mathrm{Acc}}_{00}$, we can further simplify $\widetilde{\mathrm{Acc}}$ as

$$\widetilde{\mathrm{Acc}} = \alpha\beta\left(\frac{1}{2} - \frac{1}{2}\mathrm{erf}\left(\frac{c_1 - \mu_{11}}{\sqrt{2}s}\right)\right) + (1-\alpha_1)\beta\left(\frac{1}{2} + \frac{1}{2}\mathrm{erf}\left(\frac{c_1 - \mu_{01}}{\sqrt{2}s}\right)\right)$$

$$+ \alpha_0(1-\beta)\left(\frac{1}{2} - \frac{1}{2}\mathrm{erf}\left(\frac{c_1 - \mu_{11}}{\sqrt{2}s}\right)\right) + (1-\alpha_0)(1-\beta)\left(\frac{1}{2} + \frac{1}{2}\mathrm{erf}\left(\frac{c_1 - \mu_{01}}{\sqrt{2}s}\right)\right).$$

Taking derivative of $\widetilde{\mathrm{Acc}}$ w.r.t. $c_1$, we have

$$\frac{\partial\widetilde{\mathrm{Acc}}}{\partial c_1} = -\alpha_1\beta\exp\left(-\left(\frac{c_1 - \mu_{11}}{\sqrt{2}s}\right)^2\right) + (1-\alpha_1)\beta\exp\left(-\left(\frac{c_1 - \mu_{01}}{\sqrt{2}s}\right)^2\right)$$

$$-\alpha_0(1-\beta)\exp\left(-\left(\frac{c_1 - \mu_{11}}{\sqrt{2}s}\right)^2\right) + (1-\alpha_0)(1-\beta)\exp\left(-\left(\frac{c_1 - \mu_{01}}{\sqrt{2}s}\right)^2\right).$$

Setting $\frac{\partial\widetilde{\mathrm{Acc}}}{\partial c_1} = 0$, we have

$$(\alpha_0 - \alpha_0\beta + \alpha_1\beta)\exp\left(-\left(\frac{c_1^* - \mu_{11}}{\sqrt{2}s}\right)^2\right) = (1 - \alpha_0 + \alpha_0\beta - \alpha_1\beta)\exp\left(-\left(\frac{c_1^* - \mu_{00}}{\sqrt{2}s}\right)^2\right),$$

which simplifies to $c_1^* = \frac{s^2}{\mu_{11} - \mu_{01}}\log\left(\frac{1 - \alpha_0 + \alpha_0\beta - \alpha_1\beta}{\alpha_0 - \alpha_0\beta + \alpha_1\beta}\right) + \frac{\mu_{11} + \mu_{01}}{2}$. Since $c_0^* = c_1^* + \mu_{10} - \mu_{11}$, we have $c_0^* = \frac{s^2}{\mu_{11} - \mu_{01}}\log\left(\frac{1 - \alpha_0 + \alpha_0\beta - \alpha_1\beta}{\alpha_0 - \alpha_0\beta + \alpha_1\beta}\right) + \frac{2\mu_{10} - \mu_{11} + \mu_{01}}{2}$, and $(c_0^*, c_1^*)$ gives us the optimal threshold pair under post-processing.

Accordingly, let $\mathbf{z}_b$ be the confusion vector by vanilla training, we have the change in accuracy under $(c_0^*, c_1^*)$ as

$$
\mathbf{A}_c \mathbf{z}_b - \widetilde{\mathrm{Acc}}(c_0^*, c_1^*)
$$
$$
= \mathbf{A}_c \mathbf{z}_b + \frac{1}{2} \left( \alpha_1 \beta \, \mathrm{erf} \left( \frac{c_1^* - \mu_{11}}{\sqrt{2}s} \right) - (1 - \alpha_1) \, \beta \, \mathrm{erf} \left( \frac{c_1^* - \mu_{01}}{\sqrt{2}s} \right) \right.
$$
$$
\left. + \alpha_0 (1 - \beta) \, \mathrm{erf} \left( \frac{c_0^* - \mu_{10}}{\sqrt{2}s} \right) - (1 - \alpha_0)(1 - \beta) \, \mathrm{erf} \left( \frac{c_0^* - \mu_{00}}{\sqrt{2}s} \right) \right) - \frac{1}{2}. \tag{11}
$$

Owing to the intractability of error function, we instead consider the following second-order Pade approximation (Baker Jr & Gammel, 1961) of error function for quantitative analysis, which yields a maximum absolute error of 0.013:

$$
\mathrm{erf}(x) \approx \tanh(kx),
$$

where $k = \arg\min_k \int_{-\infty}^{\infty} || \, \mathrm{erf}(x) - \tanh(kx) ||^2$.

Let $\mu_y' := k \frac{c_1^* - \mu_{y1}}{\sqrt{2}s}$ and $\gamma := \alpha_0 + \alpha_1 \beta - \alpha_0 \beta$, let $\mathbf{z}^*$ be the confusion vector of EOd optimum under group-dependent thresholding, we can rewrite Eq. equation 11 as

$$
\mathbf{A}_c \mathbf{z}_b - \mathbf{A}_c \mathbf{z}^* = \mathbf{A}_c \mathbf{z}_b - \gamma \cdot \frac{e^{\mu_1'} - e^{-\mu_1'}}{e^{\mu_1'} + e^{-\mu_1'}} + (1 - \gamma) \frac{e^{\mu_0'} - e^{-\mu_0'}}{e^{\mu_0'} + e^{-\mu_0'}} - \frac{1}{2}.
$$

which simplifies to

$$
\mathbf{A}_c \mathbf{z}_b - \mathbf{A}_c \mathbf{z}^* = \frac{2 \mathbf{A}_c \mathbf{z}_b - 1}{2} - \frac{\left( e^{2(\mu_0' + \mu_1')} - 1 \right) + (1 - 2\gamma) \left( e^{2\mu_0'} - e^{2\mu_1'} \right)}{\left( e^{2\mu_0'} + 1 \right) \left( e^{2\mu_1'} + 1 \right)}.
$$

$\square$

