# OpenReview forum: "Rethinking Pareto Frontier: On the Optimal Trade-offs in Fair Classification"
_ICLR.cc/2026/Conference — ICLR 2026 Poster_

### Official Review · Reviewer_JZwm · 2025-10-25

**Soundness:** 3
**Presentation:** 3
**Contribution:** 2
**Rating:** 4
**Confidence:** 4

**Summary:**

The paper reformulates model-specific Pareto trade-offs for fairness–accuracy and DP–EOd via convex optimization over the confusion vector, and proposes a group-dependent last-layer retraining method with theoretical guarantees and empirical gains.

**Strengths:**

Clear statement and intuitive novelty;
Multiple datasets evaluated;

**Weaknesses:**

The problem doesn’t have a single fix. The right mix of fairness and accuracy changes by case. Even on the Pareto front, the final pick depends on the use case, so it isn’t a standalone answer.

Standard fairness contraints already let you tune the balance between fairness and accuracy. So even if this study shows a better way to optimse them together, theoratically, I don‘t think it will change much in what we got after solving the problem.

People have talked about accuracy vs fairness a lot in past years. I don’t think this is the main gap now; some newer work even tries to separate the two.

**Questions:**

see above

---

> ### Author Response · Authors · 2025-11-21
>
> Thank you for taking the time and effort to review our paper.
>
> [Scope of our work] We agree with the reviewer that the appropriate fairness–accuracy trade-off is application-dependent, and our aim is not to prescribe a single operating point, but to characterize what a fixed model can achieve across the full range of fairness constraints. Our framework therefore recovers a tight, data-consistent approximation of the model-specific achievable region in confusion-vector space. For a given backbone, this yields a Pareto frontier that traces how accuracy responds as fairness constraints tighten and how different notions (e.g., DP vs EOd) conflict or align along this frontier. In contrast to standard constrained training or pre-, in-, and post-processing methods, which return isolated, intervention-dependent operating points without certifying how close they are to the best attainable trade-offs, our approach makes it possible to quantify when these methods are already near the achievable frontier and when they incur avoidable losses in either accuracy or fairness. Our last-layer retraining with group-dependent bias is designed precisely to reduce this gap in practice, and provably improves over post-processing at the EOd optimum while remaining competitive with group-thresholding.
>
> While fairness–accuracy trade-offs have been discussed extensively, prior work has largely focused on qualitative arguments, impossibility results, or specific algorithmic solutions, rather than quantitatively mapping the fairness–accuracy frontier for a deployed model. Our formulation fills this gap by turning the fairness–accuracy trade-off into a concrete auditing tool: for any trained neural network, it delineates which combinations of accuracy and fairness are attainable and which are precluded by the model itself. In practice, these frontiers provide practitioners and policymakers with a principled basis for decision-making: they can determine whether simple threshold adjustments are sufficient, whether retraining (e.g., with our last-layer variant) is required to reach a desired operating point, or whether the target specification is incompatible with the current backbone and thus necessitates more substantial changes to the model or data.

---

### Official Review · Reviewer_Nxm5 · 2025-11-03

**Soundness:** 3
**Presentation:** 3
**Contribution:** 3
**Rating:** 6
**Confidence:** 3

**Summary:**

The paper studies Pareto-optimal trade-offs in the fair classification task. It argues that prior model-specific (MS) Pareto frontiers can be suboptimal and may not reflect the true best-achievable trade-offs between accuracy and fairness, or between different fairness notions. To address this challenge, this paper reformulates MS Pareto frontiers as low-dimensional convex programs in terms of a confusion-vector parameterization, extends this to trade-offs between demographic parity and equalized odds under accuracy constraints, and proposes a last-layer retraining framework that introduces group-dependent bias terms. The theoretical comparisons and empirical results on multiple datasets validate the proposed method.

**Strengths:**

1. The problem formulation is clear. The formation from accuracy/fairness metrics to a linear form of a confusion matrix is clean and leads to tractable convex optimization.
2. The proposed research extends fairness-accuracy trade-off to a general fairness-fairness analysis with accuracy constraints.
3. The theoretical results and experimental results validate the proposed methods.

**Weaknesses:**

1. It would be promising to discuss more fairness metrics beyond EOd and DP.

**Questions:**

1. For Assumption 1: it is likely that the ROC curves of a classifier is concave. But it might be common that all group-dependent ROC curves are all concave. Even though there is a empirical validation, I wonder the explicitly conditions behind Assumption 1. Is it possible that if the majority of the group-dependent ROC curves are concave, then Lemma 1 is correct?

---

> ### Author Response · Authors · 2025-11-21
>
> Thank you for taking the time and effort to review our paper.
>
> [Weakness 1: extension to alternative fairness notions] Our formulation can be readily generalized to alternative group fairness notions. For example, the calibration-within-groups notion [1] can be formulated as
>
> $$
> \text{CG} = ||\mathbf{A}_{\text{CG}}   (\mathbf{A}_0 \mathbf{z}_1 + \mathbf{A}_1 (1-\mathbf{z}_1))||_1,
> $$
>
> where $$
> \mathbf{A}_{\text{CG}} =
> \begin{bmatrix}
>   1 - v_1 & 0 & -v_1 & 0 & 0 & 0 & 0 & 0; \\
>   0 & 1 - v_0 & 0 & -v_0 & 0 & 0 & 0 & 0; \\
>   0 & 0 & 0 & 0 & 1 - v_1 & 0 & -v_1 & 0; \\
>   0 & 0 & 0 & 0 & 0 & 1 - v_0 & 0 & -v_0
> \end{bmatrix}
> $$, $$
> \mathbf{A}_0 =
> \begin{bmatrix}
>   \alpha_0 (1-\beta) & 0 & 0 & 0; \\
>   0 & 0 & 0 & 0; \\
>   0 & 0 & 0 & 0; \\
>   0 & (1-\alpha_0)(1-\beta) & 0 & 0; \\
>   0 & 0 & \alpha_1 \beta & 0; \\
>   0 & 0 & 0 & 0; \\
>   0 & 0 & 0 & 0; \\
>   0 & 0 & 0 & (1-\alpha_1)\beta
> \end{bmatrix}
> $$, and $$
> \mathbf{A}_1 =
> \begin{bmatrix}
>   0 & 0 & 0 & 0; \\
>   \alpha_0 (1-\beta) & 0 & 0 & 0; \\
>   0 & (1-\alpha_0)(1-\beta) & 0 & 0; \\
>   0 & 0 & 0 & 0; \\
>   0 & 0 & 0 & 0; \\
>   0 & 0 & \alpha_1 \beta  & 0; \\
>   0 & 0 & 0 & (1-\alpha_1)\beta; \\
>   0 & 0 & 0 & 0
> \end{bmatrix}$$. Specifically, $\alpha_a = \text{Pr}[Y = 1|A = a]$ and $\beta = \text{Pr}[A = 1]$, as defined in line 168 of main paper, and $v\_0, v\_1$ are score representatives such that $0 \leqslant v_0, v_1 \leqslant 1$. We 'll include more discussions on alternative fairness notions in the revised paper.
>
> [Question 1: concavity assumption] Our assumption on the group-dependent ROC curves enables us to construct a **strictly tighter** achievable frontier than the one derived from Eq. 4 of the main paper. This refinement aligns more closely with the empirical behavior of neural networks, as illustrated in the following link:
>
> https://docs.google.com/document/d/1sYl-tFl-mFcn2xOMhfDh5LMCuXychpXcFy3mpjpRajU/edit?usp=sharing
>
> In practice, Lemma 1 holds whenever, for each group $a$, the group-dependent ROC curve lies above the two line segments connecting $(0,0)\rightarrow(\mathrm{FPR}_a,\mathrm{TPR}_a)$ and $(\mathrm{FPR}_a,\mathrm{TPR}_a)\rightarrow(1,1)$ in the FPR-TPR plane. This condition is weaker than strict concavity and is often satisfied even when the ROC curves are not strictly concave (as shown in the figures in the link above). Under this assumption, consider any feasible solution pair $(\mathrm{FPR}_0,\mathrm{TPR}_0)$ and $(\mathrm{FPR}_1,\mathrm{TPR}_1)$ returned by optimizing Eq. 4 of the main paper. Because each ROC curve lies above its two-segment lower envelope, the point $(\mathrm{FPR}_a,\mathrm{TPR}_a)$ for each group admits a horizontal leftward shift that keeps it within the achievable region of that group. For each group $a$, define:
>
> $$
> \delta_a = \max\\{\delta \ge 0 : (\mathrm{FPR}_a - \delta,\\, \mathrm{TPR}_a)
> \text{ lies below the ROC curve of group } a\\}.
> $$
>
> Let $\delta^\star = \min(\delta_0,\,\delta_1)$, translating both points by the same distance $\delta^\star$ yields:
>
> $$
> (\mathrm{FPR}_a,\mathrm{TPR}_a)
> \\;\longmapsto\\;
> (\mathrm{FPR}_a - \delta^\star,\\, \mathrm{TPR}_a), \qquad a \in \\{0,1\\}.
> $$
>
> This transformation preserves the EOd violation (the TPR and FPR gaps remain unchanged) while strictly reducing both FPRs, thereby improving accuracy. Thus, whenever the group-specific ROC curves lie above their respective two-segment lower envelopes, every solution produced by Eq. 4 can be strictly improved, establishing the practical sufficiency of Lemma 1.
>
> [1] Kim, Joon Sik, Jiahao Chen, and Ameet Talwalkar. "Fact: A diagnostic for group fairness trade-offs." International Conference on Machine Learning. PMLR, 2020.

---

### Official Review · Reviewer_FUh2 · 2025-11-03

**Soundness:** 3
**Presentation:** 2
**Contribution:** 3
**Rating:** 6
**Confidence:** 3

**Summary:**

This paper addresses the fundamental challenge of quantifying and optimizing trade-offs in machine learning fairness. While prior studies have observed empirical trade-offs between fairness and accuracy—and among different fairness notions—they lack a principled characterization of the best achievable (Pareto optimal) trade-offs. To fill this gap, the authors reformulate the model-specific Pareto optimal trade-off as a convex optimization problem based on the confusion vector, enabling efficient approximation of optimal accuracy under dynamic fairness constraints. Their framework also extends analysis to the trade-off between fairness notions, revealing how accuracy influences their compatibility. Building on this formulation, they propose a last-layer retraining method with group-dependent bias and provide theoretical guarantees of its superiority. Experiments validate that the proposed approach achieves improved fairness-accuracy balance and effectively quantifies both types of trade-offs.

**Strengths:**

1. The proposed low-dimensional convex program over confusion vectors is both efficient and interpretable, providing a principled way to estimate model-specific Pareto frontiers.
2. The last-layer retraining approach with group-dependent biases is easy to implement and consistently outperforms post-processing and other last-layer baselines in the experiments.
3. The paper includes solid theoretical results—such as proofs of suboptimality for prior frontiers, superiority of the proposed retraining method over random flipping, and an analytic formula for accuracy–fairness trade-offs—that reinforce the empirical findings.

**Weaknesses:**

1. Strong modeling assumptions: The theoretical results rely on restrictive assumptions (e.g., concave ROC curves and equal-variance Gaussian logits), which may not always hold in practice. The robustness of the approach under violations of these assumptions could be further investigated.
2. Dependence on $z_b$ / $z_{ab}$ estimation: The convex program depends on reliable upper-bound estimates obtained from multiple training restarts. The paper does not sufficiently analyze how sensitive the resulting frontier is to these estimates.
3. Notation overload: The manuscript uses a large number of mathematical symbols, which can make it difficult to follow. Including a concise table summarizing all symbols and variables would significantly improve readability.
4. Experimental detail and reproducibility: Some implementation details—such as the number of random initializations for $f$, the sampling strategy for the $\epsilon$-grid, and runtime or computational cost—are missing or only briefly mentioned in the appendix. Providing these would enhance transparency and reproducibility.

**Questions:**

1. How sensitive is the computed MS Pareto frontier to the estimation of $z_b$ and $z_{ab}$? Could suboptimal $z_b$ estimates (e.g., from local minima) substantially loosen the frontier? Please report the variance across multiple $z_b$ estimates.
2. For datasets or models where ROC curves are not concave, how often does Lemma 1 fail to hold? Are there diagnostic tools to detect when post-processing frontiers may already be tight?
3. The Gaussian equal-variance logit assumption in Theorem 2 is convenient but potentially unrealistic. Can you provide empirical evidence (on your datasets) showing that subgroup logit variances are comparable, or quantify how deviations affect the accuracy–drop bound?
4. What is the computational cost of the proposed approach—particularly solving the convex program for multiple $\epsilon$ samples and $z_b$ initializations? For example, how long does the process take in the ResNet-50 CelebA experiments?

---

> ### Author Response · Authors · 2025-11-21
>
> Thank you for taking the time and effort to review our paper.
>
> [Weakness 1, question 2: concavity assumption] We show the group-dependent ROC curves across different datasets in the following link:
>
> https://docs.google.com/document/d/1sYl-tFl-mFcn2xOMhfDh5LMCuXychpXcFy3mpjpRajU/edit?usp=sharing
>
> Across all three benchmarks, the group-dependent ROC curves exhibit clear empirical concavity, and this effect becomes more pronounced with larger training sets and more expressive models.
>
> In practice, Lemma 1 holds whenever, for each group $a$, the group-dependent ROC curve lies above the two line segments connecting $(0,0)\rightarrow(\mathrm{FPR}_a,\mathrm{TPR}_a)$ and $(\mathrm{FPR}_a,\mathrm{TPR}_a)\rightarrow(1,1)$ in the FPR-TPR plane. This condition is weaker than strict concavity and is often satisfied even when the ROC curves are not strictly concave (as shown in the figures in the link above). Under this assumption, consider any feasible solution pair $(\mathrm{FPR}_0,\mathrm{TPR}_0)$ and $(\mathrm{FPR}_1,\mathrm{TPR}_1)$ returned by optimizing Eq. 4 of the main paper. Because each ROC curve lies above its two-segment lower envelope, the point $(\mathrm{FPR}_a,\mathrm{TPR}_a)$ for each group admits a horizontal leftward shift that keeps it within the achievable region of that group. For each group $a$, define:
>
> $$
> \delta_a = \max\\{\delta \ge 0 : (\mathrm{FPR}_a - \delta,\\, \mathrm{TPR}_a)
> \text{ lies below the ROC curve of group } a\\}.
> $$
>
> Let $\delta^\star = \min(\delta_0,\,\delta_1)$, translating both points by the same distance $\delta^\star$ yields:
>
> $$
> (\mathrm{FPR}_a,\mathrm{TPR}_a)
> \\;\longmapsto\\;
> (\mathrm{FPR}_a - \delta^\star,\\, \mathrm{TPR}_a), \qquad a \in \\{0,1\\}.
> $$
>
> This transformation preserves the EOd violation (the TPR and FPR gaps remain unchanged) while strictly reducing both FPRs, thereby improving accuracy. Thus, whenever the group-specific ROC curves lie above their respective two-segment lower envelopes, every solution produced by Eq. 4 can be strictly improved, establishing the practical sufficiency of Lemma 1.
>
> [Weakness 1, question 2: diagnostic tools for post-processing] strictly certifying global tightness is infeasible, because it would require optimizing over all model parameters rather than just thresholds. However, we can obtain empirical evidence. First, if very small fairness gaps (e.g., ≤1%) can already be reached with only minimal accuracy reduction, then the representation likely supports a near-tight frontier for post-processing. Second, the effectiveness of post-processing depends on the learned features: if the same procedure applied to independently trained models yields strictly better frontiers, the original one was not tight; but if these frontiers remain nearly identical and stronger interventions (such as our last-layer retraining framework) bring negligible gains, this indicates that the post-processing frontier is empirically close to the best achievable for that model class.
>
> [Weakness 1, question 3: Gaussian assumption] We refer to Sec. 12 and Tab. 10 in the Supplementary for empirical validation of the accuracy–drop bound. Although our estimate may deviate slightly from the observed accuracy drops, it generally provides a reliable approximation and closely aligns with the empirical trends shown in Fig. 2 of main paper.
>
> [Weakness 2, question 1: sensitivity analysis] We show the variances across five independent estimates of $\mathbf{z}_b$ on the Adult, COMPAS and CelebA datasets in the following table:
>
> | Dataset  | Variance (\%) |
> | --------- | -------- |
> | Adult (male TPR, male TNR, female TPR, and female TNR) | \{0.17, 0.11, 0.15, 0.09\} |
> | COMPAS (Black TPR, Black TNR, White TPR, and White TNR) | \{0.15, 0.17, 0.25, 0.17\} |
> | CelebA (male TPR, male TNR, female TPR, and female TNR) | \{0.10, 0.08, 0.07, 0.13\} |
>
> As our method aggregates results from multiple random initializations, the resulting estimates are in general consistent, as reflected by the uniformly low variance.
>
>
> [Weakness 3: notation overload] Thank you for the suggestion. We include a table of notions as below:
>
> | Notation  | Meaning |
> | --------- | -------- |
> | $\alpha_a$ | Base rate of group  $a$ |
> | $\beta$ | Proportion of samples from sensitive group $a=1$ |
> | $\mathbf{z}$| Confusion vector|
> | $\mathbf{A}_{\text{c}}$| Matrix for calculating accuracy from the confusion vector |
> | $\mathbf{A}\_{\text{DP}}$, $\mathbf{A}'\_{\text{DP}}$ | Matrices for calculating DP from the confusion vector |
> | $\mathbf{A}_{\text{EOd}}$| Matrix for calculating EOd from the confusion vector |
> | $\mathbf{z}_b$| Confusion vector obtained by vanilla training |
> | $\mathbf{z}^a_b$| Confusion vector obtained by vanilla training on group $a$|
>
> We 'll include the summary table in the revised paper.

---

> ### Author Response · Authors · 2025-11-21
>
> [Weakness 4, Question 4: experimental details] We choose the number of random initializations $f$ to be 10, and sampling interval as $0.001$. We include the computational cost on CelebA dataset in the following table:
>
> | ResNet-50  | Initializations | Optimization|
> | --------- | -------- | -------- |
> | Time (s)| 3186.8    |1.09   |
>
> Most of the computational cost comes from initializing $\mathbf{z}_b$, while our optimization framework adds only a minimal overhead to the overall runtime.

---

### Official Review · Reviewer_ygdX · 2025-11-03

**Soundness:** 3
**Presentation:** 2
**Contribution:** 3
**Rating:** 6
**Confidence:** 2

**Summary:**

This paper presents a theoretically grounded and empirically validated framework for characterizing the Pareto-optimal trade-offs in fair classification, specifically focusing on the fairness-accuracy and fairness-fairness (e.g., DP vs. EOd) trade-offs. The authors identify and rigorously address the suboptimality of existing post-processing methods (e.g., FACT, G-STAR) in approximating the true Model-Specific (MS) Pareto frontier. Their core contribution is a reformulation of the problem as a convex optimization over the confusion vector, which bypasses the limitations of specific intervention algorithms. Building on this, they propose a novel last-layer retraining method with group-dependent bias and provide strong theoretical guarantees for its superiority over existing baselines. The experimental results on multiple datasets are comprehensive and convincingly support the claims.

**Strengths:**

1. The paper correctly identifies a key flaw in prior work: the use of sub-optimal fairness interventions (like post-processing with random flipping) leads to an inaccurate and pessimistic approximation of the MS Pareto frontier. The proposed reformulation using confusion vectors and convex optimization is elegant, efficient, and provides a more meaningful upper bound on achievable performance.
2. The theoretical contributions are substantial, and the empirical Validation is rich.

**Weaknesses:**

1. The entire theoretical framework in Section 4.1 relies on the assumption that group-dependent ROC curves are concave. While the authors justify this by stating that optimal classifiers are expected to have concave ROC curves, this may not hold perfectly in practice for complex, deep neural networks, especially on real-world data with noise. The paper would be strengthened by a more detailed discussion of the implications when this assumption is violated and perhaps an empirical analysis of the concavity of the ROC curves in their experiments.
2. The convex optimization for the frontier is performed over a 4-dimensional confusion vector for binary classification. The extension to multi-class (Section 13, Appendix) increases the dimensionality significantly (k classes × 2 groups). The computational cost and feasibility for large k should be briefly discussed. Is the optimization still tractable for, say, 100 classes?
3. In the multi-class extension (Equation 8), the definitions of A_EOd,k' and A_DP,k' could be explained more clearly. The current notation is somewhat dense and could benefit from a sentence or two of intuitive explanation regarding how these matrices capture the worst-case disparity across all classes.

**Questions:**

Q1: In Lemma 1 and the corresponding proof, you assume concave ROC curves. Could you provide empirical evidence in the appendix (e.g., a plot) showing the ROC curves for your models on one of the datasets to validate that they are indeed approximately concave?
Q2: The hyperparameter λ in your retraining framework (Eq. 6,7) is crucial. Could you provide more details on the sensitivity analysis? How robust is the performance to the choice of λ?
Q3: In the multi-class experiments on the Drug dataset (Table 11, 12), why are there no baselines for the DP-accuracy trade-off other than your own method? Is it because the cited works (DFR, SELF) do not target DP?

---

> ### Author Response · Authors · 2025-11-21
>
> Thank you for taking the time and effort to review our paper.
>
> [Weakness 1, Question 1: concavity assumption] We show the group-dependent ROC curves of across different datasets in the following link:
>
> https://docs.google.com/document/d/1sYl-tFl-mFcn2xOMhfDh5LMCuXychpXcFy3mpjpRajU/edit?usp=sharing
>
> Across all three benchmarks, the group-dependent ROC curves are generally concave, with this concavity becoming more pronounced as the training set size and model complexity increase. This empirical pattern supports the key assumption underlying our theoretical analysis.
>
> [Weakness 2: computational cost] The computational cost of solving Eq. 5 of main paper is governed by the dimension of the confusion vector, which is $d = \Theta(k|\mathcal{A}|)$ with $k$ classes and $|\mathcal{A}|$ sensitive groups. Using a standard SLSQP method, each solve incurs a **worst-case** cost of $O(d^3)=O\\!\big((k|\mathcal{A}|)^3\big)$ arithmetic operations per fairness level. In practice, however, the computational time is dominated by fixed Python/SciPy overhead, such as constraint callbacks and solver bookkeeping, whose costs scale only as $O(1)$-$O(d)$ and overshadow the $O(d^3)$ linear-algebra term. To illustrate this, we show the time required to generate the frontiers on CelebA and Drug datasets in the following table:
>
> |Dataset   | CelebA (4 subgroups) | Drug (8 subgroups)|
> | --------- | -------- | -------- |
> | Time (s)| 1.09    |1.45    |
>
> Increasing the number of classes from binary to four leads to less than a 40\% increase in runtime, far from the cubic growth suggested by the $O(d^3)$ bound. This validates that our method scales efficiently in practice.
>
> [Weakness 3: explanation of multi-class fairness notion] Thank you for the suggestion. We 'll include more explanations in the revised paper. Intuitively, consider a three-class classification task ($y \in \{0,1,2\}$) with binary senstive attributes ($a \in \{0,1\}$), the EOd notion ensures that across all the labels, we have
>
> $$|\text{Pr}[f(x)=y|y,a=0] - \text{Pr}[f(x)=y|y,a=1]| \leqslant \epsilon, \\, \forall y \in \\{0,1,2\\}.$$
>
> This ensures that the maximum error gap within any class between the two groups is bounded by $\epsilon$. Similarly, for the DP notion, we have
>
> $$|\text{Pr}[f(x)=y|a=0] - \text{Pr}[f(x)=y|a=1]| \leqslant \epsilon, \\, \forall y \in \\{0,1,2\\}.$$
>
> This ensures that the largest difference in the probability of predicting any class between the two groups is also bounded.
>
> [Question 2: sensitivity analysis] We include the change in EOd and DP as $\lambda$ varies in the following link:
>
> https://docs.google.com/document/d/1CCNPm0PCg_pB6HPXSSQFRXPaRtU4GKRo3fF003ppf0k/edit?usp=sharing
>
> Larger values of $\lambda$ progressively tighten the imposed fairness constraint, reducing both the EOd (left) and DP (right) gaps while inducing a gradual drop in accuracy. The EOd gap shrinks more sharply at smaller $\lambda$’s, whereas the DP gap decreases more smoothly across the range of $\lambda$ values.
>
>
> [Question 3: baseline results] Since DFR [1] and SELF [2] are designed to improve worst-group accuracy (i.e., EOd-style fairness) instead of demographic parity, we do not report their DP results to maintain fairness in comparison.
>
> [1] Kirichenko, Polina, Pavel Izmailov, and Andrew Gordon Wilson. "Last layer re-training is sufficient for robustness to spurious correlations." arXiv preprint arXiv:2204.02937 (2022).
>
> [2] LaBonte, Tyler, Vidya Muthukumar, and Abhishek Kumar. "Towards last-layer retraining for group robustness with fewer annotations." Advances in Neural Information Processing Systems 36 (2023): 11552-11579.

---

### Author Response · Authors · 2025-12-03
**Final Remarks**

Dear Area Chair,

We sincerely thank the reviewers for their constructive feedback and their recognition of our work. Our work proposes a convex-optimization based framework for reformulating model-specific Pareto frontiers, enabling a principled quantification of both (i) the fairness–accuracy trade-off and (ii) trade-offs between different fairness notions. Building upon this analysis, we further introduce a novel last-layer retraining method that provably achieves a better fairness-accuracy trade-off than prior post-processing baselines.

Following the suggestions by the reviewers, we have included more detailed analysis on the empirical validity of our key theoretical assumptions, alongside a rigorous examination of computational complexity and runtime efficiency. We have also provided comprehensive sensitivity analyses regarding hyperparameters and initialization variance, along with clarifications on notations as well as the broader scope of our framework. We summarize the key discussions below:

**1. Concavity and Gaussian Assumption**

We provided empirical evidence showing that group-dependent ROC curves exhibit **clear concavity**, which becomes more pronounced with larger training sets and model complexity. Theoretically, we clarified that a weaker condition, where ROC curves simply lie above specific linear segments, is sufficient for Lemma 1 to hold, ensuring the combination yields strictly better frontiers than the solutions derived by Eq. 4. Additionally, we validated that our Gaussian assumption for approximating accuracy drops closely aligns with empirical observations.

**2. Runtime Efficiency**

We showed that while the theoretical **worst-case** complexity of our framework is cubic with respect to the confusion vector dimension, the actual runtime is dominated by fixed overheads (e.g., solver bookkeeping). Empirical timings on the CelebA and Drug datasets demonstrate that increasing the number of subgroups or classes leads to only a marginal increase in runtime, validating that our framework scales efficiently for practical applications.

**3. Sensitivity Analysis**

We analyzed the impact of hyperparameters on our retraining method and the influence of initialization on our optimization framework.  Varying the Lagrange multiplier ($\lambda$) confirmed the expected trade-off behavior: larger values tighten fairness constraints (reducing EOd and DP gaps) at the cost of a gradual drop in accuracy. Additionally, we demonstrated that the estimated confusion vectors exhibit **low variance** across multiple random initializations, confirming the stability and reproducibility of our frontier.

**4. Scope of our Framework**

We clarified that our reformulation serves as a **quantitative auditing tool** rather than a prescriptive solution for a single operating point. By constructing the full achievable Pareto frontier for a fixed backbone, our framework enables principled decision-making regarding the necessity of interventions. Additionally, we demonstrated the generalizability of our framework to alternative fairness notions, such as calibration-within-groups.

We hope that the additional validation of our key assumptions, computational efficiency and sensitivity, helps to clarify the adaptability of our approach. We trust these discussions further elucidate the utility of our framework as a quantitative auditing tool. We will integrate these discussion into the manuscript to ensure a comprehensive presentation.


Thank you for your time and consideration.

Best,

Authors

---

### Meta-Review · Area_Chair_o2pJ · 2025-12-22

**Summary:**

Reviewers express concern about the reliance on concavity and Gaussian-logit assumptions, the lack of analysis on how violations affect performance, and limited discussion of the scalability of the multi-class extension. Some notation is heavy, and important implementation details for reproducibility are missing. Sensitivity to estimated upper bounds is not thoroughly examined. The focus on DP and EOd narrows applicability, and a few reviewers question whether improving Pareto estimation materially changes practical fairness decision-making.

**Reviewer Concerns:**

Most of the concerns have been addressed.

**Reviewer Scores:**

6,6,6,6

---

### Decision · Program_Chairs · 2026-01-26

Accept (Poster)